# Low-rank Momentum Factorization for Memory Efficient Training

**Pouria Mahdavinia**                                                    *pxm5426@psu.edu*
*Department of Computer Science and Engineering*
*The Pennsylvania State University*

**Mehrdad Mahdavi**                                                      *mzm616@psu.edu*
*Department of Computer Science and Engineering*
*The Pennsylvania State University*

**Reviewed on OpenReview:** *https://openreview.net/forum?id=W3D3TVo9a3*

## Abstract

Fine-tuning large foundation models presents significant memory challenges due to stateful optimizers like AdamW, often requiring several times more GPU memory than inference. While memory-efficient methods like parameter-efficient fine-tuning (e.g., LoRA) and optimizer state compression exist, recent approaches like GaLore bridge these by using low-rank gradient projections and subspace moment accumulation. However, such methods may struggle with fixed subspaces or computationally costly offline resampling (e.g., requiring full-matrix SVDs). We propose Momentum Factorized SGD (MoFaSGD), which maintains a dynamically updated low-rank SVD representation of the first-order momentum, closely approximating its full-rank counterpart throughout training. This factorization enables a memory-efficient fine-tuning method that adaptively updates the optimization subspace at each iteration. Crucially, MoFaSGD leverages the computed low-rank momentum factors to perform efficient spectrally normalized updates, offering an alternative to subspace moment accumulation. We establish theoretical convergence guarantees for MoFaSGD, proving it achieves an optimal rate for non-convex stochastic optimization under standard assumptions. Empirically, we demonstrate MoFaSGD's effectiveness on large language model alignment benchmarks, achieving a competitive trade-off between memory reduction (comparable to LoRA) and performance compared to state-of-the-art low-rank optimization methods. Our implementation is available at `https://github.com/pmahdavi/MoFaSGD`.

## 1 Introduction

Advancements in AI have been propelled by initial scaling laws, which boost pre-training performance via ever-larger models and datasets (Kaplan et al., 2020). However, adapting these large foundation models to downstream tasks, such as instruction or preference tuning (Ouyang et al., 2022), often requires several times more GPU memory than inference. This memory burden largely stems from storing optimizer states (e.g., momentum terms) required by ubiquitous methods like AdamW. Various strategies have emerged to alleviate these costs.

Parameter-Efficient Fine-Tuning (PEFT) methods provide an architectural solution, introducing a small set of new, trainable parameters called adapters. For example, the popular LoRA technique (Hu et al., 2021) restricts model updates to these low-rank adapters, drastically reducing memory overhead by training only a small fraction of parameters. Variants such as DoRA (Liu et al., 2024), AdaLoRA (Zhang et al., 2023), VeRA (Kopiczko et al., 2023) and rsLoRA (Kalajdzievski, 2023) have further refined this approach to low-rank parameterization. Other approaches focus on compressing optimizer states directly; methods like AdaFactor (Shazeer & Stern, 2018), SM3 (Anil et al., 2019), and quantization techniques (Modoranu

et al., 2024; Feinberg et al., 2024) factorize or reduce the precision of moments. Alternatively, stateless methods like signSGD (Bernstein et al., 2018) and SWAN (Ma et al., 2024) avoid momentum accumulation by carefully normalizing gradients each iteration.

A complementary line of research, **low-rank subspace optimization**, projects gradients onto a smaller, dynamic subspace to reduce memory. This allows full-parameter updates while operating in a lower-dimensional space. Key examples include GaLore (Zhao et al., 2024a), Flora (Hao et al., 2024), and ReLoRA (Lialin et al., 2023). However, applying these dynamic subspace techniques effectively presents difficulties.

**Challenges in Online Subspace Updates.** While promising, dynamically updating the optimization subspace online, as in GaLore, faces key obstacles. First, managing optimizer states (particularly momentum) during abrupt subspace changes is non-trivial; existing methods may reset states (Lialin et al., 2023), transform them (Hao et al., 2024), or leave them unchanged (Hao et al., 2024). Second, determining the new subspace, often based on gradients, can incur high computational costs (e.g., full SVD on large gradient matrices (Zhao et al., 2024a)), hindering scalability. These limitations highlight the need for more efficient, iteration-level subspace adaptation strategies.

**Contributions**

Motivated by the success of gradient-based techniques for memory-efficient fine-tuning, the advantages of dynamic projection subspaces over static ones, and the potential for online subspace updates to better track full-rank optimization dynamics, we address three research questions:

**(Q1)** Can we mitigate the challenges of online subspace resampling through a computationally efficient update strategy while rigorously tracking projection residuals?

**(Q2)** Can we directly estimate full-rank adaptive optimizer trajectories using low-rank factors, instead of relying on compressed accumulation of subspace moments?

**(Q3)** Can we match the theoretical per-iteration complexity of standard adaptive optimizers while maintaining LoRA-level memory savings?

We answer these questions affirmatively by introducing MoFaSGD, which maintains a dynamically updated low-rank SVD factorization of the first-order momentum. We define this factorization as:

$$\hat{\boldsymbol{M}}_t = \boldsymbol{U}_{t+1}\operatorname{Diag}(\boldsymbol{\sigma}_{t+1})\boldsymbol{V}_{t+1}^\top \approx \beta\hat{\boldsymbol{M}}_{t-1} + (1-\beta)\boldsymbol{G}_t \quad \text{(First-Momentum Factorization)} \tag{1}$$

Here, $\hat{\boldsymbol{M}}_t$ denotes the approximate first-order momentum at iteration $t$, $\boldsymbol{G}_t$ is the gradient at iteration $t$, and $\beta$ is the momentum decay factor. The low-rank SVD factors $\boldsymbol{U}_{t+1} \in \mathbb{R}^{m\times r}$ (left singular vectors), $\boldsymbol{\sigma}_{t+1} \in \mathbb{R}^r$ (singular values), and $\boldsymbol{V}_{t+1} \in \mathbb{R}^{n\times r}$ (right singular vectors) serve as the optimizer state in our algorithm.

Motivated by our observation that the exponential moving average (EMA) of gradients, defined as $\sum_{i=1}^t \beta^{t-i}\boldsymbol{G}_i$, often exhibit low-rank structure, MoFaSGD applies a specific low-rank approximation strategy: maintaining an online, low-rank SVD factorization of the *first-order momentum*. This differs from state compression methods that typically target second moments (Shazeer & Stern, 2018) and avoids the complex factored approximations of curvature used in methods like Shampoo or KFAC (Gupta et al., 2018; Martens & Grosse, 2015). The factorization is updated efficiently at each iteration using tangent space projections, bypassing costly offline resampling (Zhao et al., 2024a). Crucially, MoFaSGD leverages the *same* continuously updated low-rank momentum factors ($\boldsymbol{U}_{t+1}, \boldsymbol{V}_{t+1}$) to perform spectral normalization ($\boldsymbol{W}_{t+1} \leftarrow \boldsymbol{W}_t - \eta\boldsymbol{U}_{t+1}\boldsymbol{V}_{t+1}^\top$). This integrated approach, inspired by methods like Shampoo (Gupta et al., 2018) and Muon (Jordan et al., 2024b), provides adaptive step directions without requiring separate, computationally intensive matrix operations (e.g., SVD, matrix roots, or Newton-Schulz iterations) at each step.

MoFaSGD achieves LoRA-like memory savings through a unified low-rank momentum representation, while also enabling efficient, adaptive full-parameter updates via spectral normalization. This positions our method as a low-rank variant of Muon. Unlike MoFaSGD, Muon maintains full-rank momentum buffers ($\mathcal{O}(mn)$ memory) and applies full-rank updates after normalization.

## 2 Related Work

To improve memory efficiency for training large-scale neural networks, various strategies have emerged, including parameter-efficient fine-tuning, subspace optimization, optimizer state compression, stateless approaches, and partial update techniques. Moreover, second-order and preconditioning methods such as Shampoo (Gupta et al., 2018) have been gaining popularity due to their faster convergence compared to AdamW, albeit with even higher memory requirements, as studied in detail in Kasimbeg et al. (2025).

Below, we detail relevant methods like subspace optimization, optimizer state compression, stateless approaches, and gradient spectral normalization; other approaches are deferred to Appendix A.

**Subspace Optimization**

Rather than introducing additional parameters through PEFT adapters, subspace optimization methods focus on reducing the memory footprint of gradients and optimizer states directly by projecting gradients onto a low-rank subspace and performing moment accumulation in the subspace as well (Zhao et al., 2024a; Gur-Ari et al., 2018; Gressmann et al., 2020; Yang et al., 2023; Vogels et al., 2020). GaLore (Zhao et al., 2024a) projects gradients onto a subspace defined by either the left or right singular vectors of the gradients, computed via SVD, and accumulates optimizer states within this subspace, thereby reducing the required optimizer memory. Flora (Hao et al., 2024) periodically resamples the subspace projection matrix using a multivariate Gaussian distribution for low-rank gradient projections.

AdaRankGrad (Refael et al., 2024) dynamically adjusts gradient rank during training. LDAdam (Robert et al., 2024) and APOLLO (Zhu et al., 2024) maintain optimizer states in low-dimensional representations, adopting similar strategy to subspace moment accumulation as in GaLore (Zhao et al., 2024a). Subspace management is a key challenge. Computing it can require costly operations like full SVD (Zhao et al., 2024a), and its update frequency creates a trade-off: infrequent updates can lead to stale information, while frequent updates can disrupt optimizer state accumulation. Managing optimizer states across subspace changes also requires careful strategies (Lialin et al., 2023; Hao et al., 2024).

**Optimizer State Compression.** This approach directly reduces the memory footprint of existing optimizer states, primarily the second moments used in adaptive methods. AdaFactor (Shazeer & Stern, 2018) and SM3 (Anil et al., 2019) achieve this through factorization techniques, reducing memory complexity from $\mathcal{O}(mn)$ to $\mathcal{O}(m+n)$. Quantization methods like 8-bit Adam (Dettmers et al., 2021) and 4-bit variants (Li et al., 2023) reduce the precision of stored moments, achieving near full-precision performance with significant memory savings. CAME (Luo et al., 2023) and Adapprox (Zhao et al., 2024b) refine these compression techniques for potentially better accuracy.

**Stateless and Gradient Normalization Methods.** Eliminating optimizer states entirely provides maximal memory efficiency. signSGD (Bernstein et al., 2018) achieves this by using only the sign of the gradient. Lion (Chen et al., 2024) similarly reduces the memory footprint of AdamW by discarding the second-moment estimate. Gradient normalization methods such as SWAN (Ma et al., 2024) and Muon (Jordan et al., 2024b) emulate the behavior of adaptive optimizers without storing second-moment statistics. Muon, in particular, applies momentum followed by approximate orthogonalization of the momentum matrix ($\boldsymbol{U_M V_M^\top}$, where $\boldsymbol{M = U_M \Sigma_M V_M^\top}$) using efficient Newton-Schulz iterations (Jordan et al., 2024b), drawing connections to accumulation-free variants of Shampoo (Jordan et al., 2024b; Bernstein & Newhouse, 2024b). However, Muon still retains the Nesterov momentum, and thus is not considered fully stateless.

**Positioning MoFaSGD.** The landscape of memory-efficient optimization presents a spectrum of trade-offs. Architectural-based solutions such as PEFT methods offer high memory savings but restrict fine-tuning updates (Hu et al., 2021) to a low-rank space. Subspace methods enable full-rank parameter updates but encounter challenges such as costly or potentially disruptive subspace resampling and moment accumulation (Zhao et al., 2024a; Hao et al., 2024); moreover, subspace techniques are mostly applied to AdamW, while their application to non-diagonal-based preconditioning methods (such as Shampoo and its variants (Gupta et al., 2018; George et al., 2018; Jordan et al., 2024b)) remains largely unexplored. Optimizer state compression techniques focus on reducing the storage cost of second moments (Shazeer & Stern, 2018; Dettmers

et al., 2021). Stateless methods are memory-efficient but discard historical information (Bernstein et al., 2018; Ma et al., 2024). This work introduces MoFaSGD to navigate these challenges.

## 3 Preliminaries

This section introduces notation and provides background on subspace and adaptive optimization methods.

**Notation and Conventions.** We denote matrices with bold capital letters (e.g., $\boldsymbol{X}$), vectors with bold lowercase letters (e.g., $\boldsymbol{x}$), and scalars with non-bold letters (e.g., $x$). The Frobenius inner product for matrices is denoted as $\langle \cdot, \cdot \rangle$ (e.g., $\langle \boldsymbol{X}, \boldsymbol{Y} \rangle = \mathrm{Tr}(\boldsymbol{X}^\top \boldsymbol{Y})$). Norms are denoted by $\| \cdot \|$, with specific norms indicated using subscripts (e.g., $\| \cdot \|_{\mathrm{F}}$ for the Frobenius norm). The loss function is represented by $\mathcal{L}(\cdot)$. Without loss of generality, we denote the model parameters as $\boldsymbol{W}_t \in \mathbb{R}^{m \times n}$, and the full-rank gradient at $\boldsymbol{W}_t$ is denoted as $\boldsymbol{G}_t = \frac{\partial \mathcal{L}(\boldsymbol{W})}{\partial \boldsymbol{W}}|_{\boldsymbol{W}_t} \in \mathbb{R}^{m \times n}$. The subscript index $t$ denotes the iteration step of any optimizer for each weight matrix. Gradients are defined based on the specific batches used in the corresponding iteration. To simplify notation, we use the same symbols as previously defined where applicable. For any matrix $\boldsymbol{X} \in \mathbb{R}^{m_1 \times m_2}$, the vectorization operator $\mathrm{Vec}(\cdot)$ stacks its columns into a single vector, denoted as $\boldsymbol{x} = \mathrm{Vec}(\boldsymbol{X}) \in \mathbb{R}^{m_1 m_2 \times 1}$. The vectorized version of a matrix is always denoted by the corresponding bold lowercase letter. The Kronecker product is denoted as $\otimes$, such that for $\boldsymbol{X} \in \mathbb{R}^{m_1 \times n_1}$ and $\boldsymbol{Y} \in \mathbb{R}^{m_2 \times n_2}$, the Kronecker product $\boldsymbol{X} \otimes \boldsymbol{Y} \in \mathbb{R}^{m_1 m_2 \times n_1 n_2}$. The symbol $\odot$ denotes the element-wise (Hadamard) product between two matrices of the same dimensions. See Appendix D.2 for further notational details.

### Adaptive methods with Switched-off Momentums

Without first-order momentum, adaptive methods can be framed as a form of online mirror descent:

$$\boldsymbol{w}_{t+1} = \arg\min_{\boldsymbol{w}} \mathcal{L}(\boldsymbol{w}_t) + \langle \boldsymbol{w} - \boldsymbol{w}_t, \boldsymbol{g}_t \rangle + \frac{1}{2\eta}\|\boldsymbol{w} - \boldsymbol{w}_t\|_{\boldsymbol{P}_t} = \boldsymbol{w}_t - \eta \boldsymbol{P}_t^{-1} \boldsymbol{g}_t \tag{2}$$

where the preconditioner $\boldsymbol{P}_t$ can be any positive semidefinite matrix, and $\mathcal{L}(\boldsymbol{w}_t) + \langle \boldsymbol{w}_{t+1} - \boldsymbol{w}_t, \boldsymbol{g}_t \rangle$ is the local approximation of the loss function. For instance, if we let $\boldsymbol{P}_{t,\mathrm{AdaGrad}} = \left[\sum_{i=1}^t \boldsymbol{g}_i \boldsymbol{g}_i^\top\right]^{\frac{1}{2}}$, we exactly recover full-matrix Adagrad (Duchi et al., 2011).

Similarly, if we define $\boldsymbol{L}_t = \sum_{i=1}^t \boldsymbol{G}_i \boldsymbol{G}_i^\top$ and $\boldsymbol{R}_t = \sum_{i=1}^t \boldsymbol{G}_i^\top \boldsymbol{G}_i$, then letting $\boldsymbol{P}_{t,\mathrm{Shampoo}} = (\boldsymbol{R}_t \otimes \boldsymbol{L}_t)^{\frac{1}{4}}$ recovers the Shampoo updates for matrices (Gupta et al., 2018). Furthermore, if we consider that $\boldsymbol{g}_t \boldsymbol{g}_t^\top$ closely approximates the Gauss-Newton components of the true Hessian (Morwani et al., 2024), this motivates switching off covariance momentum (second moment) and considering the following preconditioners based on Adagrad and Shampoo: $\boldsymbol{P}_{t,1} = \left[\boldsymbol{g}_t \boldsymbol{g}_t^\top\right]^{\frac{1}{2}}$ and $\boldsymbol{P}_{t,2} = (\boldsymbol{G}_t^\top \boldsymbol{G}_t \otimes \boldsymbol{G}_t \boldsymbol{G}_t^\top)^{\frac{1}{4}}$.

A simple derivation shows that using the diagonal version of $\boldsymbol{P}_{t,1}$ in Equation 2 recovers Signed-SGD (Bernstein et al., 2018). Moreover, if we consider the reduced SVD of the gradient $\boldsymbol{G}_t = \boldsymbol{U}_{\boldsymbol{G}_t} \boldsymbol{\Sigma}_{\boldsymbol{G}_t} \boldsymbol{V}_{\boldsymbol{G}_t}^\top$, then we can write $\boldsymbol{P}_{t,2} = (\boldsymbol{V}_{\boldsymbol{G}_t} \boldsymbol{\Sigma}_{\boldsymbol{G}_t}^2 \boldsymbol{V}_{\boldsymbol{G}_t}^\top \otimes \boldsymbol{U}_{\boldsymbol{G}_t} \boldsymbol{\Sigma}_{\boldsymbol{G}_t}^2 \boldsymbol{U}_{\boldsymbol{G}_t}^\top)^{\frac{1}{4}}$. Using Lemma D.1 in the appendix, we have:

$$\boldsymbol{P}_{t,2} = (\boldsymbol{V}_{\boldsymbol{G}_t} \boldsymbol{\Sigma}_{\boldsymbol{G}_t}^2 \boldsymbol{V}_{\boldsymbol{G}_t}^\top \otimes \boldsymbol{U}_{\boldsymbol{G}_t} \boldsymbol{\Sigma}_{\boldsymbol{G}_t}^2 \boldsymbol{U}_{\boldsymbol{G}_t}^\top)^{\frac{1}{4}} = (\boldsymbol{V}_{\boldsymbol{G}_t} \otimes \boldsymbol{U}_{\boldsymbol{G}_t})(\boldsymbol{\Sigma}_{\boldsymbol{G}_t}^{\frac{1}{2}} \otimes \boldsymbol{\Sigma}_{\boldsymbol{G}_t}^{\frac{1}{2}})(\boldsymbol{V}_{\boldsymbol{G}_t} \otimes \boldsymbol{U}_{\boldsymbol{G}_t})^\top \tag{3}$$

Furthermore, we can write $\boldsymbol{g}_t = \mathrm{Vec}(\boldsymbol{G}_t) = (\boldsymbol{V}_{\boldsymbol{G}_t} \otimes \boldsymbol{U}_{\boldsymbol{G}_t}) \mathrm{Vec}(\boldsymbol{\Sigma}_{\boldsymbol{G}_t})$. Substituting the preconditioner $\boldsymbol{P}_{t,2}$ and the vectorized gradient into Equation 2 yields:

$$\begin{aligned} \boldsymbol{w}_{t+1} &= \boldsymbol{w}_t - \eta (\boldsymbol{V}_{\boldsymbol{G}_t} \otimes \boldsymbol{U}_{\boldsymbol{G}_t})(\boldsymbol{\Sigma}_{\boldsymbol{G}_t}^{-\frac{1}{2}} \otimes \boldsymbol{\Sigma}_{\boldsymbol{G}_t}^{-\frac{1}{2}})(\boldsymbol{V}_{\boldsymbol{G}_t} \otimes \boldsymbol{U}_{\boldsymbol{G}_t})^\top \boldsymbol{g}_t \\ &= \boldsymbol{w}_t - \eta (\boldsymbol{V}_{\boldsymbol{G}_t} \otimes \boldsymbol{U}_{\boldsymbol{G}_t})(\boldsymbol{\Sigma}_{\boldsymbol{G}_t}^{-\frac{1}{2}} \otimes \boldsymbol{\Sigma}_{\boldsymbol{G}_t}^{-\frac{1}{2}}) \mathrm{Vec}(\boldsymbol{\Sigma}_{\boldsymbol{G}_t}) \\ &= \boldsymbol{w}_t - \eta (\boldsymbol{V}_{\boldsymbol{G}_t} \otimes \boldsymbol{U}_{\boldsymbol{G}_t}) \mathrm{Vec}(\boldsymbol{I}_r) = \boldsymbol{w}_t - \eta \boldsymbol{U}_{\boldsymbol{G}_t} \boldsymbol{V}_{\boldsymbol{G}_t}^\top \end{aligned} \tag{4}$$

Thus, using $\boldsymbol{P}_{t,2}$ recovers spectrally normalized updates, sometimes referred to as gradient whitening (Jordan et al., 2024b; Ma et al., 2024). For a similar derivation, see Bernstein & Newhouse (2024b). Additional background on the historical development of adaptive optimization methods is provided in Appendix B.

**Subspace Optimization Methods**

Many studies support the conjecture that gradients during the training of deep neural networks exhibit a low-rank structure, lying in a low-dimensional subspace (Gur-Ari et al., 2018; Gressmann et al., 2020; Yang et al., 2023). This property has been studied both theoretically and empirically and has been leveraged to improve optimization algorithms ranging from communication-efficient distributed training (Vogels et al., 2020) to efficient fine-tuning (Hu et al., 2021; Lialin et al., 2023; Hao et al., 2024; Zhao et al., 2024a). Zhao et al. (2024a) aim to explicitly leverage this property to perform subspace training, where gradients are projected and accumulated in a low-rank subspace, leading to significant memory footprint reduction.

We briefly clarify subspace methods mathematically, following the formulation of GaLore (Zhao et al., 2024a). Let $\boldsymbol{W}_t \in \mathbb{R}^{m \times n}$ represent the model parameters, and let $\boldsymbol{Q}_t \in \mathbb{R}^{m \times r}$ be the projection matrix defining the subspace at iteration $t$. Then, GaLore performs the following update:

$$
\begin{aligned}
\boldsymbol{G}_{t,r} &= \boldsymbol{Q}_t^\top \boldsymbol{G}_t \in \mathbb{R}^{r \times n} && \text{(Subspace projection)} \\
\boldsymbol{M}_{t,r} &= \beta_1 \boldsymbol{M}_{t-1,r} + (1-\beta_1)\boldsymbol{G}_{t,r} && \text{(First subspace moment)} \\
\boldsymbol{V}_{t,r} &= \beta_2 \boldsymbol{V}_{t-1,r} + (1-\beta_2)(\boldsymbol{G}_{t,r} \odot \boldsymbol{G}_{t,r}) && \text{(Second subspace moment)} \\
\boldsymbol{W}_{t+1} &= \boldsymbol{W}_t - \eta_t \boldsymbol{Q}_t \left( \frac{\boldsymbol{M}_{t,r}}{\sqrt{\boldsymbol{V}_{t,r}}} \right) && \text{(Project back and update)}
\end{aligned}
\tag{5}
$$

Moreover, by switching off subspace momentum accumulations, the GaLore update can simply be seen as projected gradient descent: $\boldsymbol{W}_{t+1} = \boldsymbol{W}_t - \eta_t \boldsymbol{Q}_t \boldsymbol{Q}_t^\top \boldsymbol{G}_t$. In addition to subspace accumulation, GaLore performs offline updates of the subspace $\boldsymbol{Q}_t$ by updating it as $\boldsymbol{U}_{\boldsymbol{G}_t}^{1:r}$, the top-$r$ left singular vectors of $\boldsymbol{G}_t$, obtained via an SVD operation at predetermined intervals.

## 4 Momentum Factorized SGD with Spectral Normalization

This section introduces MoFaSGD, our memory-efficient optimization method, and its theoretical underpinnings. We begin in Subsection 4.1 by detailing the core algorithmic ideas behind MoFaSGD. This includes the motivation derived from the low-rank structure observed in optimizers, the process of maintaining and efficiently updating low-rank momentum factors using tangent space projection, and the subsequent use of these factors for spectrally normalized parameter updates (Algorithm 1). Following the algorithmic description, Subsection 4.2 presents the convergence properties and theoretical analysis of MoFaSGD, justifying our design choices and establishing formal performance guarantees.

### 4.1 The MoFaSGD Algorithm

**Motivation: Low-Rank Momentum Factors**

Feinberg et al. (2024) show that the EMA of gradient covariance $\sum_{i=1}^t \beta^{t-i}\boldsymbol{G}_i\boldsymbol{G}_i^\top$ and $\sum_{i=1}^t \beta^{t-i}\boldsymbol{G}_i^\top\boldsymbol{G}_i$ maintain low-rank structure and spectral decay properties throughout training, and Zhao et al. (2024a) argue that the gradients themselves become low-rank during fine-tuning. Building on these observations, we conjecture that the gradient EMAs exhibit low-rank properties. We experimentally evaluate this conjecture in Section 5.3 and show that the mass of the gradient EMA is largely centered on its top few singular values. Low-rank structure has also been widely leveraged for LLM fine-tuning, e.g., LoRA (Hu et al., 2021) adapts low-rank matrices during training, while GaLore (Zhao et al., 2024a) leverages the low-rank structure of gradients. Thus, leveraging the low-rank property of the gradient EMA connects the implicit assumptions underlying GaLore and LoRA.

Building on this conjecture, we propose to maintain a low-rank SVD factorized representation of the first momentum. We highlight that this factorization plays two major roles in our algorithm design. First, it is leveraged for online subspace sampling, and second, it provides a low-rank estimation of the first momentum (gradient EMA). Formally, we define the low-rank moment factors as:

$$
\hat{\boldsymbol{M}}_t \triangleq \boldsymbol{U}_{t+1}\boldsymbol{\Sigma}_{t+1}\boldsymbol{V}_{t+1}^\top \approx \sum_{i=1}^t \beta^{t-i}\boldsymbol{G}_i \qquad \text{(Low-rank Moment Factors)}
$$

---

**Algorithm 1 MoFaSGD**: Momentum Factorized Stochastic Gradient Descent

---

**Require:** Step size $\eta$, decay $\beta$, rank $r$
**Ensure:** Optimized weights $\boldsymbol{W}_t$

1: **Initialize:** $\boldsymbol{W}_0$
2: $\boldsymbol{G}_0 \leftarrow \nabla_{\boldsymbol{W}} \mathcal{L}(\boldsymbol{W}_0)$
3: Initialize moment factors: $(\boldsymbol{U}_0, \boldsymbol{\Sigma}_0, \boldsymbol{V}_0) \leftarrow \text{SVD}_r(\boldsymbol{G}_0)$
4: $t \leftarrow 0$
5: **repeat**
6: $\quad \boldsymbol{G}_t \leftarrow \nabla_{\boldsymbol{W}} \mathcal{L}(\boldsymbol{W}_t)$
7: $\quad (\boldsymbol{U}_{t+1}, \boldsymbol{\Sigma}_{t+1}, \boldsymbol{V}_{t+1}) \leftarrow \text{UMF}(\boldsymbol{G}_t, \boldsymbol{U}_t, \boldsymbol{\Sigma}_t, \boldsymbol{V}_t, \beta)$
8: $\quad \boldsymbol{W}_{t+1} \leftarrow \boldsymbol{W}_t - \eta \boldsymbol{U}_{t+1} \boldsymbol{V}_{t+1}^\top$
9: $\quad t \leftarrow t + 1$
10: **until** convergence criterion is met
11: **return** $\boldsymbol{W}_t$

**function** UMF($\boldsymbol{G}_t, \boldsymbol{U}_t, \boldsymbol{\Sigma}_t, \boldsymbol{V}_t, \beta$)

1: *Compute subspace projections:*
2: $\quad \boldsymbol{G}_t \boldsymbol{V}_t, \boldsymbol{U}_t^\top \boldsymbol{G}_t, \boldsymbol{U}_t^\top \boldsymbol{G}_t \boldsymbol{V}_t$
3: Compute QR factors:
4: $\quad (\boldsymbol{U}_t', \boldsymbol{R}_{\boldsymbol{U}_t}) = \text{QR}([\boldsymbol{U}_t \quad \boldsymbol{G}_t \boldsymbol{V}_t])$
5: $\quad (\boldsymbol{V}_t', \boldsymbol{R}_{\boldsymbol{V}_t}) = \text{QR}([\boldsymbol{V}_t \quad \boldsymbol{G}_t^\top \boldsymbol{U}_t])$
6: Construct $2r \times 2r$ matrix:
7: $\quad \boldsymbol{S}_t = \boldsymbol{R}_{\boldsymbol{U}_t} \begin{bmatrix} \beta\boldsymbol{\Sigma}_t - \boldsymbol{U}_t^\top \boldsymbol{G}_t \boldsymbol{V}_t & \boldsymbol{I}_r \\ \boldsymbol{I}_r & \boldsymbol{0}_r \end{bmatrix} \boldsymbol{R}_{\boldsymbol{V}_t}^\top$
8: Compute rank-$r$ SVD:
9: $\quad \boldsymbol{U}_t'' \boldsymbol{\Sigma}_t'' (\boldsymbol{V}_t'')^\top \leftarrow \text{SVD}_r(\boldsymbol{S}_t)$
10: $\boldsymbol{U}_{t+1} \leftarrow \boldsymbol{U}_t' \boldsymbol{U}_t''$
11: $\boldsymbol{V}_{t+1} \leftarrow \boldsymbol{V}_t' \boldsymbol{V}_t''$
12: $\boldsymbol{\Sigma}_{t+1} \leftarrow \boldsymbol{\Sigma}_t''$
13: **return** $(\boldsymbol{U}_{t+1}, \boldsymbol{\Sigma}_{t+1}, \boldsymbol{V}_{t+1})$

---

where $\{\boldsymbol{G}_i\}_{i=1}^t$ are the observed gradients until iteration $t$. The left moment factor $\boldsymbol{U}_{t+1} \in \mathbb{R}^{m \times r}$ and the right moment factor $\boldsymbol{V}_{t+1} \in \mathbb{R}^{n \times r}$ are orthogonal matrices, while $\boldsymbol{\Sigma}_{t+1} \in \mathbb{R}^{r \times r}$ is diagonal.

Our method maps the full-rank gradient $\boldsymbol{G}_t$ to the tangent space of the previous moment factor representation $(\boldsymbol{U}_t, \boldsymbol{\Sigma}_t, \boldsymbol{V}_t)$, to ensure a smooth adaptation of the subspace when a new gradient arrives. Formally, we leverage the tangent space of the previous iteration $\mathcal{T}_{(\boldsymbol{U}_t, \boldsymbol{\Sigma}_t, \boldsymbol{V}_t)}$ as the new subspace for projection, which we denote as $\mathcal{T}_t$:

$$\mathcal{T}_t = \{\boldsymbol{U}_t \boldsymbol{M} \boldsymbol{V}_t^\top + \boldsymbol{U}_p \boldsymbol{V}_t^\top + \boldsymbol{U}_t \boldsymbol{V}_p^\top | \boldsymbol{M} \in \mathbb{R}^{r \times r}, \boldsymbol{U}_p \in \mathbb{R}^{m \times r}, \boldsymbol{V}_p \in \mathbb{R}^{n \times r}, \boldsymbol{U}_t^\top \boldsymbol{U}_p = \boldsymbol{0}, \boldsymbol{V}_t^\top \boldsymbol{V}_p = \boldsymbol{0}\} \quad (6)$$

and the projection to this subspace can be derived as follows:

$$\hat{\boldsymbol{G}}_t \triangleq \text{Proj}_{\mathcal{T}_t}(\boldsymbol{G}_t) = \boldsymbol{U}_t \boldsymbol{U}_t^\top \boldsymbol{G}_t + \boldsymbol{G}_t \boldsymbol{V}_t \boldsymbol{V}_t^\top - \boldsymbol{U}_t \boldsymbol{U}_t^\top \boldsymbol{G}_t \boldsymbol{V}_t \boldsymbol{V}_t^\top \qquad \text{(Online subspace projection)}$$

where the definition of projection to the tangent space is $\text{Proj}_{\mathcal{T}_t}(\boldsymbol{G}_t) = \underset{\boldsymbol{G} \in \mathcal{T}_t}{\arg\min} \|\boldsymbol{G} - \boldsymbol{G}_t\|_\text{F}$.

Projecting onto the tangent subspace of previous gradients as shown in Theorem 4.3, results in a lower compression error compared to the left, right, or two-sided subspace projections used in GaLore (Zhao et al., 2024a).

**Efficient Momentum Factor Updates**

The second component of our approach is to efficiently approximate the current moment factor $\hat{\boldsymbol{M}}_t$ given the projected gradient $\hat{\boldsymbol{G}}_t$ and the previous moment factor $\hat{\boldsymbol{M}}_{t-1}$. A naive update, $\hat{\boldsymbol{M}}_t = \text{SVD}_r(\hat{\boldsymbol{G}}_t + \beta\hat{\boldsymbol{M}}_{t-1})$ where $\text{SVD}_r(\cdot)$ denotes the rank-$r$ truncated SVD, involves a computationally expensive SVD operation, which we aim to avoid. Since both $\hat{\boldsymbol{M}}_{t-1}$ and $\hat{\boldsymbol{G}}_t$ are rank-$r$, their sum has a rank of at most $2r$. This observation allows us to approximate $\hat{\boldsymbol{M}}_t$ in $\mathcal{O}((m+n)r^2)$, which is far more efficient than a full-matrix SVD. Let $(\boldsymbol{U}_t', \boldsymbol{R}_{\boldsymbol{U}_t}) = \text{QR}([\boldsymbol{U}_t \quad \boldsymbol{G}_t \boldsymbol{V}_t])$, and $(\boldsymbol{V}_t', \boldsymbol{R}_{\boldsymbol{V}_t}) = \text{QR}([\boldsymbol{V}_t \quad \boldsymbol{G}_t^\top \boldsymbol{U}_t])$, where QR stands for the QR decomposition, and $\boldsymbol{U}_t' \in \mathbb{R}^{m \times 2r}$, $\boldsymbol{V}_t' \in \mathbb{R}^{n \times 2r}$ and $\boldsymbol{R}_{\boldsymbol{U}_t}, \boldsymbol{R}_{\boldsymbol{V}_t} \in \mathbb{R}^{2r \times 2r}$. Then we can write:

$$\hat{\boldsymbol{G}}_t + \beta\hat{\boldsymbol{M}}_{t-1} = \boldsymbol{U}_t \boldsymbol{U}_t^\top \boldsymbol{G}_t + \boldsymbol{G}_t \boldsymbol{V}_t \boldsymbol{V}_t^\top + \boldsymbol{U}_t(\beta\boldsymbol{\Sigma}_t - \boldsymbol{U}_t^\top \boldsymbol{G}_t \boldsymbol{V}_t)\boldsymbol{V}_t^\top$$
$$= [\boldsymbol{U}_t \quad \boldsymbol{G}_t \boldsymbol{V}_t] \begin{bmatrix} \beta\boldsymbol{\Sigma}_t - \boldsymbol{U}_t^\top \boldsymbol{G}_t \boldsymbol{V}_t & \boldsymbol{I}_r \\ \boldsymbol{I}_r & \boldsymbol{0}_r \end{bmatrix} \begin{bmatrix} \boldsymbol{V}_t^\top \\ \boldsymbol{U}_t^\top \boldsymbol{G}_t \end{bmatrix} = \boldsymbol{U}_t'\left(\boldsymbol{R}_{\boldsymbol{U}_t} \begin{bmatrix} \beta\boldsymbol{\Sigma}_t - \boldsymbol{U}_t^\top \boldsymbol{G}_t \boldsymbol{V}_t & \boldsymbol{I}_r \\ \boldsymbol{I}_r & \boldsymbol{0}_r \end{bmatrix} \boldsymbol{R}_{\boldsymbol{V}_t}^\top\right)\boldsymbol{V}_t'^\top \quad (7)$$

Let $\boldsymbol{U}_t'' \boldsymbol{\Sigma}_t'' \boldsymbol{V}_t''^\top = \text{SVD}_r\left(\boldsymbol{R}_{\boldsymbol{U}_t} \begin{bmatrix} \beta\boldsymbol{\Sigma}_t - \boldsymbol{U}_t^\top \boldsymbol{G}_t \boldsymbol{V}_t & \boldsymbol{I}_r \\ \boldsymbol{I}_r & \boldsymbol{0}_r \end{bmatrix} \boldsymbol{R}_{\boldsymbol{V}_t}^\top\right)$, and note that the inner matrix has rank at most $r$, and $\boldsymbol{U}_t'', \boldsymbol{V}_t'' \in \mathbb{R}^{2r \times r}$. We can finally write our momentum factor update rule as:

$$\boldsymbol{U}_{t+1} = \boldsymbol{U}_t' \boldsymbol{U}_t'' \quad , \quad \boldsymbol{V}_{t+1} = \boldsymbol{V}_t' \boldsymbol{V}_t'' \quad , \quad \boldsymbol{\Sigma}_{t+1} = \boldsymbol{\Sigma}_t'' \quad (8)$$

The computation complexity of our approach includes two QR decompositions, $\mathcal{O}((m+n)r^2)$, one full SVD on a $2r \times 2r$ matrix, $\mathcal{O}(r^3)$, and hence the total complexity is $\mathcal{O}((m+n)r^2 + r^3)$.

**From Momentum Factors to Spectrally Normalized Updates**

Inspired by the connection between spectrally normalized gradient updates and effective non-diagonal preconditioning methods like Shampoo (Gupta et al., 2018), and motivated by the strong empirical performance of Muon (Jordan et al., 2024b), which applies spectral normalization to gradient momentum, we leverage our low-rank momentum factorization (Equation Low-rank Moment Factors) for the main optimizer step. The MoFaSGD update rule is:

$$\boldsymbol{W}_{t+1} = \boldsymbol{W}_t - \eta \boldsymbol{U}_{t+1} \boldsymbol{V}_{t+1}^\top \tag{9}$$

Here, $\boldsymbol{U}_{t+1} \in \mathbb{R}^{m \times r}$ and $\boldsymbol{V}_{t+1} \in \mathbb{R}^{n \times r}$ represent the left and right singular vectors derived from the efficiently computed low-rank approximation of the first-order momentum, $\hat{\boldsymbol{M}}_t$. MoFaSGD (summarized in Algorithm 1) contrasts with Muon, which operates on the full-rank momentum $\boldsymbol{M}_t$ (requiring $\mathcal{O}(mn)$ memory) and uses Newton-Schulz iterations to approximate $\boldsymbol{U}_{\boldsymbol{M}_t} \boldsymbol{V}_{\boldsymbol{M}_t}^\top$ for its update step. MoFaSGD can thus be viewed as a memory-efficient, low-rank variant of Muon.

Furthermore, we highlight the key distinctions between MoFaSGD and GaLore (Zhao et al., 2024a). GaLore employs a two-stage process: 1) Projecting gradients onto a low-rank subspace defined by the singular vectors of the *gradient* itself, updated periodically (referred to as offline subspace resampling), and 2) Accumulating first and second moments (akin to Adam) within this low-rank subspace.

MoFaSGD adopts different strategies for both stages. Firstly, regarding the subspace definition, MoFaSGD performs gradient low-rank projection, but crucially, onto the *tangent space* defined by the singular vectors of the *gradient momentum* ($\hat{\boldsymbol{M}}_t$). This online, per-iteration subspace adaptation contrasts with GaLore's offline resampling based on single gradients. The choice of the tangent space projection is theoretically motivated by its optimality in minimizing projection residuals (Theorem 4.3), while using the momentum's singular vectors aims for a more stable subspace that evolves smoothly, mitigating potential noise in individual gradients.

Secondly, concerning the optimizer update, MoFaSGD deliberately avoids accumulating moments *within* the subspace, unlike GaLore, thereby avoiding potential error propagation from stale subspaces. MoFaSGD directly uses the computed momentum factors ($\boldsymbol{U}_{t+1}, \boldsymbol{V}_{t+1}$) to perform the spectrally normalized update in Equation 9. This design choice aims to circumvent potential errors arising from subspace moment accumulation, particularly when the subspace changes frequently (i.e., near-online updates, or small subspace update intervals in GaLore). As empirically supported in Section 5.3, frequent subspace updates can indeed negatively impact GaLore's performance, suggesting that subspace moment accumulation errors might increase with the frequency of subspace changes.

MoFaSGD's *novelty* lies in its unique combination of: 1) Projecting gradients onto the dynamically updated tangent space derived from the low-rank *momentum* factors, and 2) Utilizing these factors directly for spectrally normalized updates, thereby bypassing the potential pitfalls of subspace moment accumulation inherent in methods like GaLore.

## 4.2 Convergence and Theoretical Analysis

Our analysis addresses the non-convex optimization problem:

$$\min_{\boldsymbol{W} \in \mathbb{R}^{m \times n}} \mathcal{L}(\boldsymbol{W}) = \mathbb{E}_\xi \big[ \mathcal{L}(\boldsymbol{W}, \xi) \big] \tag{10}$$

where we assume access to an unbiased, variance-bounded stochastic gradient oracle $\nabla \mathcal{L}(\boldsymbol{W}, \xi)$. Below, we first introduce the necessary definitions and assumptions that underpin our theoretical results.

**Definitions and Assumptions**

For any optimization iterate $\boldsymbol{W}_i$, we denote the full-batch gradient by $\bar{\boldsymbol{G}}_i = \nabla \mathcal{L}(\boldsymbol{W}_i)$ and the stochastic gradient by $\boldsymbol{G}_i = \nabla \mathcal{L}(\boldsymbol{W}_i, \xi_i)$. Formally, we leverage the following standard assumptions throughout our analysis:

**Assumption 4.1.** *$\mathcal{L}(.)$ is $L$-smooth with respect to the nuclear norm $\|.\|_*$. In other words, for any two arbitrary $\boldsymbol{W}_1, \boldsymbol{W}_2 \in \mathbb{R}^{m \times n}$, we have: $\|\nabla\mathcal{L}(\boldsymbol{W}_1) - \nabla\mathcal{L}(\boldsymbol{W}_2)\|_* \leq L\|\boldsymbol{W}_1 - \boldsymbol{W}_2\|_2$*

This assumption naturally generalizes the typical smoothness condition from vector optimization and has been previously utilized in the literature (Large et al., 2025; Bernstein & Newhouse, 2024b). For further details, please see Appendix D.1. Additionally, we assume the availability of a stochastic gradient oracle satisfying standard properties:

**Assumption 4.2.** *For any model parameter $\boldsymbol{W}$, we have access to an unbiased and variance-bounded stochastic oracle, as follows: $\mathbb{E}_\xi[\nabla\mathcal{L}(\boldsymbol{W}, \xi)] = \nabla\mathcal{L}(\boldsymbol{W})$, and $\mathbb{E}_\xi[\|\nabla\mathcal{L}(\boldsymbol{W}, \xi) - \nabla\mathcal{L}(\boldsymbol{W})\|_*] \leq \sigma$*

With these definitions and assumptions clarified, we now present an intuitive overview of our main theoretical results, which highlight the strengths and optimality of our proposed MoFaSGD algorithm.

**Optimality of Tangent Space Projection (Theorem 4.3).** We establish that projecting each gradient $\boldsymbol{G}_t$ onto the tangent space defined by its singular vectors achieves the minimal projection residual error among a broad class of low-rank projection schemes, such as projection onto the left or right singular vector subspaces. **Optimal $O(1/\sqrt{T})$ Convergence Rate (Theorem 4.5).** Under Assumptions 4.1 and 4.2, MoFaSGD achieves convergence to a stationary point at the optimal $O(1/\sqrt{T})$ rate. Critically, the factorization of momentum does not degrade the asymptotic convergence rate.

### Proof Outline

Our proof structure involves three primary steps. First, we decompose the momentum low-rank approximation error by defining the full-rank momentum as $\boldsymbol{M}_t = \sum_{i=0}^t \beta^{t-i}\boldsymbol{G}_i$, and breaking down the approximation error into two key components: $\|\hat{\boldsymbol{M}}_t - \bar{\boldsymbol{G}}_t\|_* \leq \|\boldsymbol{M}_t - \bar{\boldsymbol{G}}_t\|_* + \|\hat{\boldsymbol{M}}_t - \boldsymbol{M}_t\|_*$. These terms are individually controlled via exponential averaging and tangent-space projections. Second, we rigorously prove the optimality of the tangent-space projection (Theorem 4.3), demonstrating that $\hat{\boldsymbol{G}}_t = \boldsymbol{U}_t\boldsymbol{U}_t^\top\boldsymbol{G}_t + \boldsymbol{G}_t\boldsymbol{V}_t\boldsymbol{V}_t^\top - \boldsymbol{U}_t\boldsymbol{U}_t^\top\boldsymbol{G}_t\boldsymbol{V}_t\boldsymbol{V}_t^\top$ minimizes the residual $\|\boldsymbol{G}_t - \hat{\boldsymbol{G}}_t\|_F$, thus ensuring optimal fitting into the evolving momentum subspace. Lastly, by combining the upper bounds on the aforementioned terms with a standard descent lemma under nuclear-norm smoothness, we derive our final convergence behavior. By carefully bounding the approximation terms from the prior steps and leveraging them in our derived descent lemma under nuclear-norm smoothness, we arrive at our main convergence result.

The low-rank factorization of gradient momentum as described by Equation Low-rank Moment Factors is the cornerstone of our proposed method. The quality of the momentum approximation plays a key role in the effectiveness of our approach. We provide an intuitive sketch of the theoretical analysis to bound the factorization residual $\|\hat{\boldsymbol{M}}_t - \boldsymbol{M}_t\|_* = \|\sum_{i=1}^t \beta^{t-i}\boldsymbol{G}_i - \boldsymbol{U}_{t+1}\boldsymbol{\Sigma}_{t+1}\boldsymbol{V}_{t+1}^\top\|_F^2$. By recursively bounding this term, we can show that it is sufficient to bound the term $\|\boldsymbol{U}_{t+1}\boldsymbol{\Sigma}_{t+1}\boldsymbol{V}_{t+1}^\top - \beta\boldsymbol{U}_t\boldsymbol{\Sigma}_t\boldsymbol{V}^\top - \boldsymbol{G}_t\|_F$, which can itself be shown to be bounded by $\|\operatorname{Proj}_{\mathcal{T}_t}(\boldsymbol{G}_t) - \boldsymbol{G}_t\|_F$. Thus, the quality of the factored momentum approximation is directly related to the residual of the gradient's low-rank projection.

### Results

The choice of tangent space projection is optimal in the sense of minimizing the term $\|\operatorname{Proj}_{\mathcal{T}_t}(\boldsymbol{G}_t) - \boldsymbol{G}_t\|_F$, as demonstrated in the following theorem.

**Theorem 4.3.** *Let $\boldsymbol{L} \in \mathbb{R}^{m \times r}$, $\boldsymbol{R} \in \mathbb{R}^{n \times r}$ be any arbitrary sketching matrices, and $(\alpha_1, \alpha_2, \alpha_3)$ be any arbitrary triple of scalars. Let $\operatorname{Proj}_{(\boldsymbol{L}, \boldsymbol{R})}(\boldsymbol{G}) = \alpha_1\boldsymbol{L}\boldsymbol{L}^\top\boldsymbol{G} + \alpha_2\boldsymbol{G}\boldsymbol{R}\boldsymbol{R}^\top + \alpha_3\boldsymbol{L}\boldsymbol{L}^\top\boldsymbol{G}\boldsymbol{R}\boldsymbol{R}^\top$. Then, the projection residual is minimized when $(1)(\alpha_1, \alpha_2, \alpha_3) = (1, 1, -1)$ and $(2)\boldsymbol{L}^\top\boldsymbol{L} = \boldsymbol{R}^\top\boldsymbol{R} = \boldsymbol{I}_r$. In this case, the residual norm is $\|\operatorname{Proj}_{(\boldsymbol{L}, \boldsymbol{R})}(\boldsymbol{G}_t) - \boldsymbol{G}_t\| = \|(\boldsymbol{I} - \boldsymbol{L}\boldsymbol{L}^\top)\boldsymbol{G}_t(\boldsymbol{I} - \boldsymbol{R}\boldsymbol{R}^\top)\|$.*

*Remark* 4.4. If we let $\boldsymbol{L} = \boldsymbol{U}_{\boldsymbol{G}_t}^{1:r}$ and $\boldsymbol{R} = \boldsymbol{V}_{\boldsymbol{G}_t}^{1:r}$, then we can conclude that the residual error would be upper bounded by $\sigma_{\boldsymbol{G}_t}^{r+1:\min(m,n)} = \sum_{i=r+1}^{\min(m,n)} \sigma_{\boldsymbol{G}_t}^i$, which, considering the low-rank property of the gradient, we expect it to be small, or even zero if the rank of the full gradient is less than $r$. Note that the subspace projection in GaLore (Zhao et al., 2024a) is actually equivalent to letting either $(\alpha_1, \alpha_2, \alpha_3) = (0, 0, 1)$ or $(\alpha_1, \alpha_2, \alpha_3) = (1, 0, 0)$.

We now present our main convergence bound for Algorithm 1.

**Theorem 4.5.** *Let Assumptions 4.1 and 4.2 hold. Moreover, assume* $\operatorname{rank}(\boldsymbol{G}_0) \leq r$. *By letting* $\beta \leq \frac{1}{3}$ *and* $\eta \leq 1$, *the iterates of Algorithm 1 satisfy the following:*

$$\frac{1}{T}\sum_{t=0}^{T}\mathbb{E}[\|\nabla\mathcal{L}(\boldsymbol{W}_t)\|_*] \leq \mathcal{O}\left(\frac{\mathcal{L}(\boldsymbol{W}_0) - \mathbb{E}[\mathcal{L}(\boldsymbol{W}_{T+1})]}{\eta T} + \eta L + \frac{\sigma}{\sqrt{T}}\right) \tag{11}$$

*Moreover, if we set* $\eta = \Theta\left(\sqrt{\frac{\mathcal{L}(\boldsymbol{W}_0) - \mathbb{E}[\mathcal{L}(\boldsymbol{W}_{T+1})]}{TL}}\right)$, *we can derive the following simplified bound as:*

$$\frac{1}{T}\sum_{t=0}^{T}\mathbb{E}[\|\nabla\mathcal{L}(\boldsymbol{W}_t)\|_*] \leq \mathcal{O}\left(\frac{\left(\mathcal{L}(\boldsymbol{W}_0) - \mathbb{E}[\mathcal{L}(\boldsymbol{W}_{T+1})]\right)^{\frac{1}{2}}\sqrt{L} + \sigma}{\sqrt{T}}\right) \tag{12}$$

*Remark* 4.6. The convergence metric used for Theorem 4.5 is the stationary point with respect to the nuclear norm, which is averaged over all gradients. When setting $\sigma = 0$, Algorithm 1 achieves the rate of $\mathcal{O}(\frac{1}{\sqrt{T}})$, which is known to be optimal in the sense of non-convex stochastic optimization under smoothness and an unbiased, bounded stochastic gradient oracle (Arjevani et al., 2023).

## 5 Experiments

We evaluate MoFaSGD's effectiveness and efficiency across three large language modeling setups: pre-training, natural language understanding (NLU) fine-tuning, and instruction-tuning. These setups allow us to assess MoFaSGD's performance across different training regimes and task complexities.

### 5.1 Pre-training setup: NanoGPT Speedrun

We first evaluate MoFaSGD in a pre-training context using the Modded NanoGPT benchmark (Jordan et al., 2024a). This benchmark focuses on training a GPT-2 architecture on a subset of the FineWeb dataset (Penedo et al., 2025) and measures performance using validation perplexity on a held-out partition of FineWeb. The benchmark's default optimizer is Muon (Jordan et al., 2024b), which holds current training speed records for this task and serves as a strong, competitive baseline.

We compare MoFaSGD against full fine-tuning baselines AdamW (Loshchilov et al., 2017) and Muon, as well as the low-rank baseline GaLore (Zhao et al., 2024a). Following the standard NanoGPT speedrun setup, we use the hyperparameters tuned for Muon. We tune the learning rates for AdamW, GaLore, and MoFaSGD, along with GaLore's SVD frequency and MoFaSGD's momentum decay ($\beta$) via grid search, selecting the best configuration based on final validation perplexity (details in Appendix C.2). We evaluate ranks $r \in \{16, 32, 128\}$, common choices for low-rank methods in pre-training setup.

Our primary experiment uses a budget of 0.73 billion tokens from FineWeb, aligning with the budget used by Muon to reach the target perplexity of 3.27. Figure 3a shows the validation perplexity curves. MoFaSGD consistently outperforms GaLore across all tested ranks, achieving lower final perplexity. The performance advantage is particularly noticeable at lower ranks ($r = 16$). This suggests MoFaSGD's dynamic subspace tracking is more effective at capturing important momentum directions than GaLore's infrequent updates, especially under strict rank constraints. Both low-rank methods underperform the full-rank AdamW and Muon baselines within this specific token budget. This is likely because full-rank methods have more degrees of freedom, and the $0.73B$ token budget, optimized for Muon's convergence speed, might be insufficient for low-rank methods to fully match their performance.

To assess longer-term performance, we conduct an extended run for $10,000$ steps ($\sim 5.3B$ tokens) using rank $r = 32$ for both MoFaSGD and GaLore. As shown in Figure 3b, MoFaSGD maintains its performance advantage over GaLore, indicating that the benefits of its momentum factorization approach persist during longer training phases.

**Ablation: Convergence vs. Efficiency**

We conduct a detailed ablation study on the effect of the low-rank parameter $r \in \{16, 32, 128\}$ during NanoGPT pre-training for both GaLore and MoFaSGD. As illustrated in Figure 1 and Figure 2, higher ranks consistently improve convergence speed and final validation loss. MoFaSGD achieves smoother loss curves and stronger performance across all ranks, particularly under tight memory budgets (e.g., $r = 16$), where GaLore exhibits noticeable instability. This supports our claim that MoFaSGD's tangent-space subspace tracking better preserves optimizer continuity under aggressive compression.

In terms of runtime (Table 1), GaLore shows minimal runtime variation across ranks due to dominant offline SVD costs. In contrast, MoFaSGD's runtime scales more significantly with rank, reflecting its per-step online factorization cost. Nonetheless, MoFaSGD remains faster than GaLore at $r = 32$ and achieves superior final loss, highlighting a favorable trade-off between expressivity and computational efficiency.

Table 1: Comparison of MoFaSGD and GaLore across different ranks during NanoGPT pre-training. Best values per row are in **bold**.

| Rank | Final Val Loss | | Runtime (s) | | Throughput | |
|------|---------|--------|---------|--------|----------|----------|
| | MoFaSGD | GaLore | MoFaSGD | GaLore | MoFaSGD | GaLore |
| 16 | **3.8981** | 4.0773 | **2156** | 3755 | **338,450** | 194,395 |
| 32 | **3.7208** | 3.8953 | 3972 | **3911** | 183,770 | **186,619** |
| 128 | **3.5700** | 3.6561 | 4817 | **3839** | 151,527 | **190,131** |

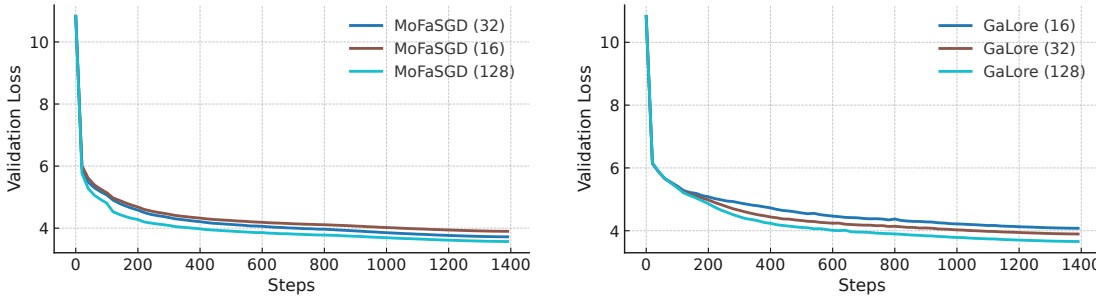

Figure 1: Validation loss vs. training steps for MoFaSGD (left) and GaLore (right) across ranks $r \in \{16, 32, 128\}$. MoFaSGD shows smoother and faster convergence.

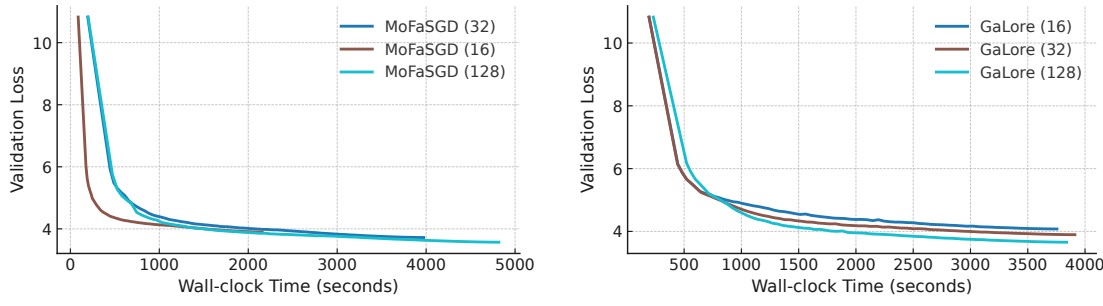

Figure 2: Validation loss vs. wall-clock time across ranks. MoFaSGD scales better in convergence and runtime efficiency.

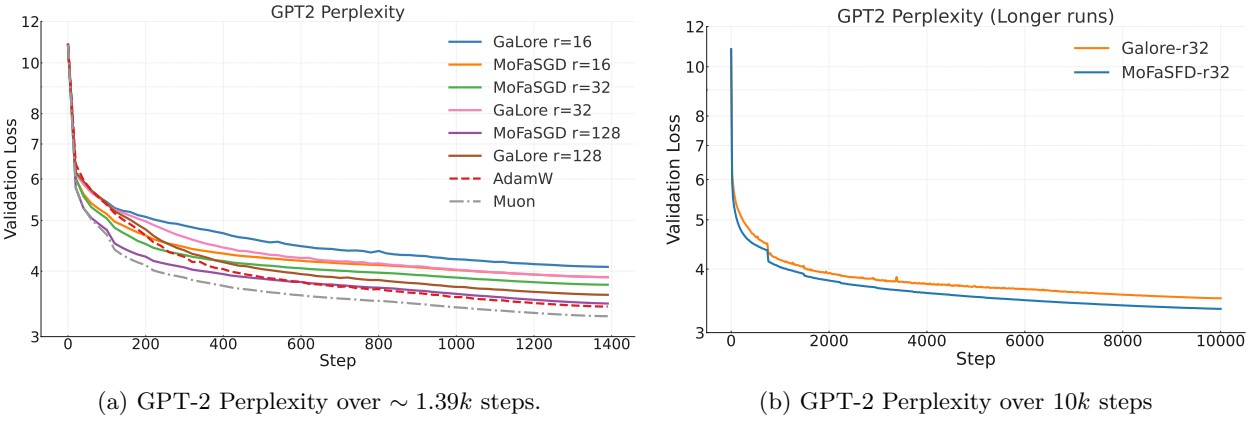

(a) GPT-2 Perplexity over $\sim 1.39k$ steps.

(b) GPT-2 Perplexity over $10k$ steps

Figure 3: Comparison of optimizer performance on GPT-2 using validation perplexity loss.

## 5.2 Post-training Evaluation: Setups and Results

We evaluate MoFaSGD in two post-training scenarios: standard NLU fine-tuning on the GLUE benchmark and large-scale instruction-tuning on the Tulu3 dataset.

**GLUE Benchmark.** We follow the experimental setup detailed in Zhao et al. (2024a) for fine-tuning a RoBERTa-Base model ($125M$ parameters) on seven diverse NLU tasks from the GLUE benchmark (Wang, 2018): MNLI, QQP, SST-2, MRPC, COLA, QNLI, and RTE. We use the same hyperparameters as Zhao et al. (2024a) for comparison baselines to ensure fairness. For MoFaSGD (with ranks $r = 4$ and $r = 8$), we tune the learning rate for each task based on validation accuracy, keeping the momentum decay $\beta$ fixed at 0.95 (details in Appendix C.3). Final validation accuracy and loss are used for evaluation.

**Tulu3 Instruction Tuning.** To assess performance on more complex alignment tasks, we fine-tune the LLaMA-3.1 8B model on the `tulu-3-sft-mixture` dataset, a large ($\sim 900K$ samples) and diverse instruction-tuning dataset (Lambert et al., 2024). We adopt most hyperparameters from the Tulu3 setup (Lambert et al., 2024), training for one epoch with an effective batch size of 128; however, we use only a subsample of 200K examples to reduce the training budget. We perform a grid search over learning rates for each optimizer. For the low-rank methods (MoFaSGD, LoRA, GaLore), we use rank $r = 8$. Key hyperparameters for MoFaSGD ($\beta$) and the baselines (GaLore SVD frequency, LoRA alpha) are set as specified in Appendix C.4. We evaluate the final checkpoints using the OLMES evaluation framework (Gu et al., 2024), benchmarking across MMLU, TruthfulQA, BigBenchHard, GSM8K, and HumanEval.

### Performance Analysis

In these post-training tasks, we compare MoFaSGD against LoRA (Hu et al., 2021) optimized with AdamW, and GaLore (Zhao et al., 2024a), which are prevalent methods for memory-efficient fine-tuning. We also include results for full-parameter fine-tuning with AdamW as a performance ceiling reference. Table 2 compares the theoretical complexities, highlighting MoFaSGD's comparable memory footprint to LoRA alongside efficient online subspace updates, while Figure 4 comprehensively details the memory usage of MoFaSGD compared to other baselines in our Tulu3 instruction-tuning setup.

To illustrate MoFaSGD's optimization efficiency during instruction tuning on the Tulu3 benchmark, we compare its validation loss trajectory against GaLore and LoRA across both training epochs and wall-clock time. As shown in Figure 5a and Figure 5b, MoFaSGD consistently achieves lower validation loss over the course of training, indicating superior sample efficiency. Notably, when measured against real-world wall-clock time, MoFaSGD converges faster than both GaLore and LoRA, demonstrating improved practical efficiency in addition to theoretical gains. These trends are further supported by our throughput analysis: MoFaSGD reaches 4206 tokens/sec, outperforming GaLore (3214 tokens/sec) and approaching the high throughput of LoRA (4536 tokens/sec). These findings reinforce our earlier conclusion that MoFaSGD's

| Optimizer | Memory Complexity | Subspace Resampling |
|---|---|---|
| GaLore | $mn + mr + 2nr$ | $\mathcal{O}(m^2 n)$ (offline) |
| LoRA | $mn + 3mr + 3nr$ | $-$ |
| MoFaSGD | $mn + mr + nr + r$ | $\mathcal{O}((m+n)r^3)$ (online) |

Table 2: Comparison of memory and subspace resampling complexity for low-rank optimizers. Let $W \in \mathbb{R}^{m \times n}$ represent model parameters, and $r$ is the rank of low-rank optimizers (w.l.o.g assume $m \le n$). Note that memory complexity includes model parameters and optimizer states.

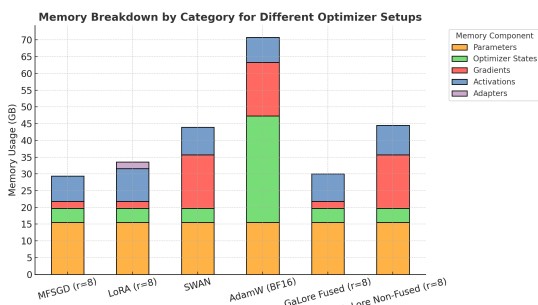

Figure 4: Empirical memory breakdown (GB) for LLaMA3.1-8B using different optimizers.

spectrally normalized updates and dynamic momentum factorization enable more effective fine-tuning under memory constraints. We have also included the training loss curves in Appendix C.5, showcasing MoFaSGD's convergence behavior on both GLUE and Tulu3 setups.

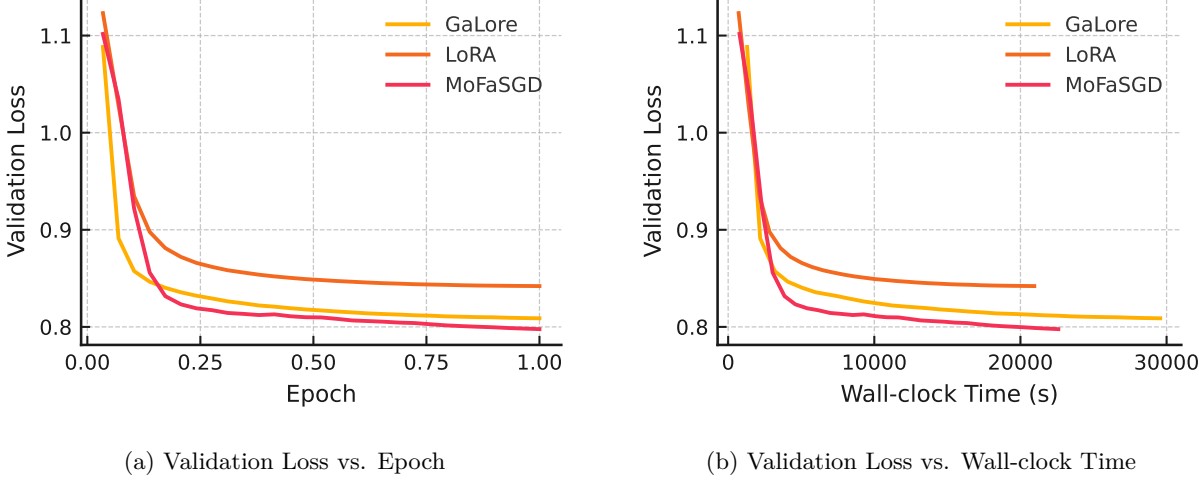

(a) Validation Loss vs. Epoch

(b) Validation Loss vs. Wall-clock Time

Figure 5: MoFaSGD demonstrates superior sample efficiency and faster wall-clock convergence, achieving lower validation loss than both GaLore and LoRA throughout LLaMA3.1-8B instruction tuning on the Tulu3 benchmark.

Table 3 summarizes the final validation accuracies on the GLUE tasks (ranks r=4 and r=8). MoFaSGD achieves performance comparable to, and on average slightly better ($+0.5\%$ for $r = 4$, $+0.47\%$ for $r = 8$) than both LoRA and GaLore, while using slightly less estimated memory. This demonstrates MoFaSGD's competitiveness and memory efficiency on standard NLU benchmarks.

Table 4 presents the results on the challenging instruction-tuning benchmarks using the Tulu3 setup (rank $r = 8$). MoFaSGD outperforms both GaLore ($+0.8\%$ avg.) and LoRA ($+2.3\%$ avg.) on the average score across the five benchmarks. This highlights MoFaSGD's potential advantage in complex tasks where accurately capturing the training dynamics over long sequences and diverse instructions is crucial. However, consistent with prior work (Wang et al., 2023), all low-rank methods exhibit a performance gap compared to full fine-tuning with AdamW (MoFaSGD is $-4.2\%$ avg. vs AdamW). This gap underscores the inherent trade-off between memory efficiency and achievable performance, particularly as task complexity increases and may require capturing subtle, high-rank parameter updates that low-rank approximations inherently miss.

Table 3: Comparison of final validation accuracies (%) on seven GLUE tasks (MNLI, QQP, SST-2, MRPC, CoLA, QNLI, RTE) when fine-tuning a RoBERTa-base model using different optimizers. For GaLore, LoRA, and MoFaSGD, we report results with rank $r \in \{4, 8\}$. Memory usage is estimated for each method, and the final column shows the average accuracy. Note that for memory measurement, we include only the parameters and the optimizer states for a fair comparison.

| Optimizer | MNLI | QQP | SST-2 | MRPC | CoLA | QNLI | RTE | Memory | Avg. |
|---|---|---|---|---|---|---|---|---|---|
| AdamW (Full-Rank) | 86.8 | 91.98 | 94.48 | 90.90 | 62.25 | 93.15 | 79.41 | 747M | 85.57 |
| GaLore ($r = 4$) | **85.23** | 89.62 | **94.17** | 90.72 | 60.33 | **93.20** | 77.22 | 253M | 84.36 |
| LoRA ($r = 4$) | 84.25 | 89.73 | 93.59 | 90.53 | 60.42 | 92.91 | 78.59 | 257M | 84.29 |
| MoFaSGD ($r = 4$) | 85.12 | **89.85** | 94.15 | **90.78** | **61.91** | 93.10 | **79.08** | 251M | 84.86 |
| GaLore ($r = 8$) | 86.01 | 89.65 | 94.04 | 90.65 | 59.96 | **93.16** | 78.14 | 257M | 84.52 |
| LoRA ($r = 8$) | 85.18 | 90.15 | 93.87 | **90.82** | 61.11 | 93.05 | 78.77 | 264M | 84.71 |
| MoFaSGD ($r = 8$) | **86.32** | **90.26** | **94.36** | 90.75 | **62.16** | 93.12 | **79.28** | 253M | 85.18 |

Table 4: Final scores of Llama-3.1 8B on the Tulu3-SFT-mixture dataset using four different optimizers. The table reports performance on MMLU, TruthfulQA, BigBenchHard, GSM8K, and HumanEval, along with the average of these five benchmarks (Avg.).

| Optimizer | MMLU | TruthfulQA | BigBenchHard | GSM8K | HumanEval | Avg. |
|---|---|---|---|---|---|---|
| AdamW (Full-Rank) | 62.8 | 46.5 | 66.7 | 72.7 | 81.0 | 65.9 |
| GaLore | 58.9 | 44.2 | 57.6 | 68.4 | 75.2 | 60.9 |
| LoRA | 56.1 | 42.6 | 56.5 | 67.9 | 73.9 | 59.4 |
| MoFaSGD | **59.4** | **45.8** | **58.3** | **68.4** | **76.8** | **61.7** |

## 5.3 Ablations

### Momentum Spectral Analysis

MoFaSGD is motivated by the conjecture that the first moment (the EMA of gradients) preserves a low-rank structure throughout training. This conjecture stems from the GaLore hypothesis on the low-rankness of the gradients themselves (Zhao et al., 2024a), and, more importantly, from the observation in Feinberg et al. (2024), which shows a fast spectral decay in the EMA of the gradient covariance, as discussed in more detail in Section 4.1. To investigate our conjecture, we analyze the first-moment buffer $M_t$ from the AdamW optimizer states generated during the Tulu3 instruction-tuning setup.

For each relevant parameter matrix $M_t$, we perform SVD and compute the energy ratio captured by the top-$r$ singular values as $\frac{\sum_{i=1}^{r} \sigma_{i,M_t}^2}{\|M_t\|_{\mathrm{F}}^2}$. This ratio represents the percentage of the momentum's total energy contained within the top-$r$ subspace. We compute the average of this ratio across all 2D weight matrices in the model at various training steps. Figure 6a shows the average energy ratio for $r = 16$ and $r = 32$ throughout training. We observe that the top-32 singular values consistently capture around 80% of the momentum's energy, while the top-16 capture approximately 75%. This persistent and significant concentration of energy in a low-rank subspace strongly supports our hypothesis.

### Impact of GaLore's Subspace Update Frequency

We analyze the impact of subspace update frequency in methods like GaLore compared to MoFaSGD's implicit per-iteration adaptation. MoFaSGD updates its momentum factors at every step, while GaLore performs explicit, costly subspace updates (e.g., full SVD on the gradient) at intervals $\tau$. To investigate whether simply increasing GaLore's update frequency (decreasing $\tau$) matches the benefits of MoFaSGD's online subspace adaptation, we conduct an ablation study on $\tau$ using the NanoGPT pre-training setup (

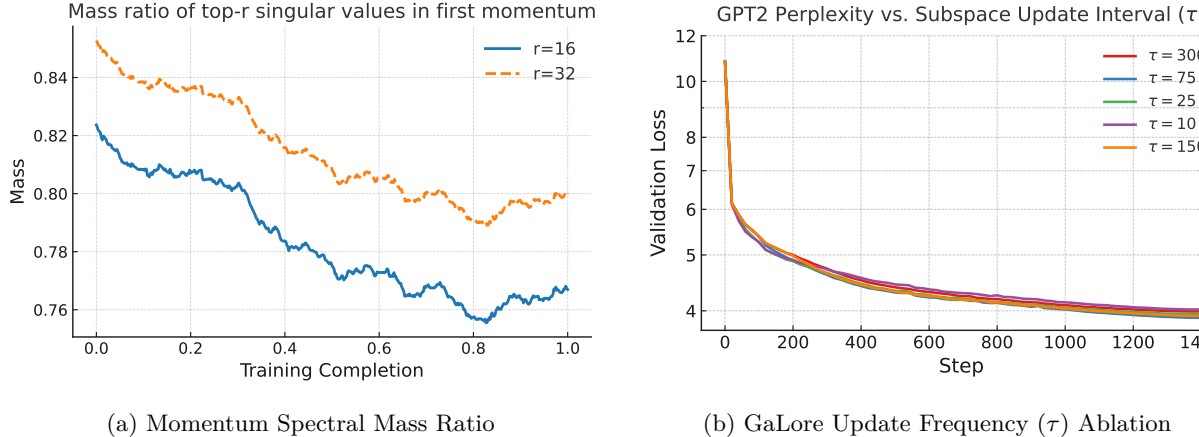

(a) Momentum Spectral Mass Ratio                    (b) GaLore Update Frequency ($\tau$) Ablation

Figure 6: (a) Average mass ratio of the AdamW first moment captured by top-$r$ singular vectors during Tulu3 fine-tuning ($r = 16, 32$). (b) Validation perplexity vs. subspace update interval ($\tau$) for GaLore ($r = 32$) on NanoGPT.

$0.73B$ token budget with rank $r = 32$). We vary $\tau$ across $\{10, 25, 75, 150, 300\}$ steps. Figure 6b shows the validation perplexity curves.

Very frequent updates ($\tau = 10$ or $\tau = 25$) do not yield the best performance and are slightly worse than less frequent updates (e.g., $\tau = 150$). This aligns with GaLore's findings (Zhao et al., 2024a) and suggests that overly frequent subspace changes in GaLore can disrupt optimizer state accumulation. This finding underscores the challenge of online subspace changes. This finding suggests that MoFaSGD's approach of avoiding abrupt subspace changes offers a more stable and efficient way to leverage low-rank structures.

## 5.4 Memory Usage Breakdown and Profiling

We assess MoFaSGD's memory efficiency by decomposing its GPU memory usage across five categories: parameters, optimizer states, gradients, activations, and adapters. Figure 4 shows a comparative breakdown of memory consumption across six optimizer setups on LLaMA3.1-8B. MoFaSGD achieves total memory usage of 29.4 GB, competitive with fused GaLore and LoRA, while enabling full-parameter updates. In contrast, AdamW exceeds 70 GB due to high-cost full-rank momentum buffers and persistent gradient accumulation. SWAN and GaLore (non-fused) similarly suffer from gradient buffers, which dominate their memory footprints.

These savings arise from three key design elements: (i) eliminating second-moment buffers entirely, (ii) maintaining a low-rank SVD factorization of first-order momentum, and (iii) fusing gradient projection and zeroing operations during backpropagation, which prevents gradient accumulation from persisting across steps.

To further validate these results, Figure 7 shows the memory trace during MoFaSGD training. We observe a clean separation between parameter storage, a narrow and persistent optimizer state band, and tightly bounded gradient memory. Compared to AdamW (Appendix C.6), which shows 16 GB persistent gradient buffers and 32 GB optimizer states, MoFaSGD significantly reduces runtime memory pressure. A full quantitative table with GB-level memory usage across all optimizers is provided in Appendix C.6.

## 5.5 Implementation Details

**Initialization.** To initialize the momentum factors in MoFaSGD, we perform a full singular value decomposition (SVD) once at the beginning of training, using the gradient from the first step. Specifically, we set $\boldsymbol{U}_0 = \boldsymbol{U}_{\boldsymbol{G}_0}^{1:r}$, $\boldsymbol{V}_0 = \boldsymbol{V}_{\boldsymbol{G}_0}^{1:r}$, and $\boldsymbol{\Sigma}_0 = \boldsymbol{\Sigma}_{\boldsymbol{G}_0}^{1:r}$, corresponding to the top-$r$ components. We apply MoFaSGD exclusively to the linear layers of transformer blocks, as spectral normalization is particularly effective for

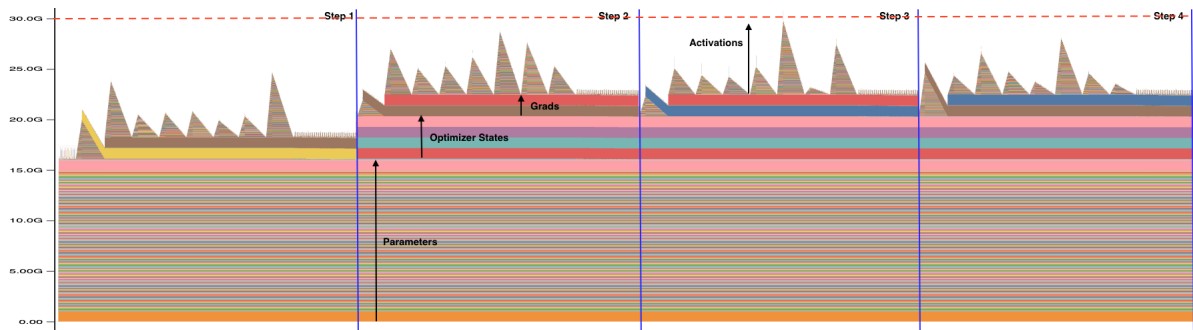

Figure 7: GPU memory trace during LLaMA3.1-8B training with MoFaSGD ($r = 8$).

these layers. Notably, both GaLore (Zhao et al., 2024a) and Muon (Jordan et al., 2024b) adopt similar approach: applying their custom optimizers only to transformer linear layers while using AdamW for embedding weights and 1D layers. We follow this convention and use AdamW (in `bf16`) for those remaining layers. Consequently, the GPU memory usage for optimizer states—including in fused-GaLore and MoFaSGD—is approximately $4.2$ GB.

**Gradient Accumulation and Fused Implementation**

Gradient accumulation is essential for training large models with limited hardware memory. For methods such as GaLore (Zhao et al., 2024a) and MoFaSGD, promptly clearing gradient buffers via backward hooks is crucial; otherwise, memory savings akin to LoRA are not realized (see Figure 14). To address this, GaLore performs a fused optimizer step, where the gradient is used to update the corresponding parameter immediately during the backward pass, after which the gradient buffer is cleared.

However, this approach is incompatible with gradient accumulation over multiple micro-batches, since the optimizer step must occur only after all gradients are accumulated. To resolve this, we introduce low-rank gradient buffers into the optimizer state and register a backward hook that accumulates low-rank projected gradients instead of performing the optimizer step immediately. For MoFaSGD, updating the momentum factors only requires the low-rank projections $G_t V_t$, $U_t^\top G_t$, and $U_t^\top G_t V_t$. We thus implement a temporary low-rank gradient buffer, enabling gradient accumulation while avoiding the need for a persistent full-rank buffer (see memory usage in Figure 7).

A similar strategy is used for GaLore: we implement a gradient-accumulation-friendly fused version by storing full-rank gradients in low-rank form. Since GaLore only needs the projection $Q_t^\top G_t$ to update its subspace momentum, low-rank accumulation suffices.

**Stateless optimizers**

Recent stateless optimizers such as SWAN (Ma et al., 2024) and SinkGD (Scetbon et al., 2025) currently do not have open-source implementations, and we encountered challenges in achieving stable convergence with our preliminary implementations. Nevertheless, given the structural similarity between SWAN and Muon (Jordan et al., 2024b) without momentum, we approximate the memory usage of such approaches by profiling Muon with its momentum buffer disabled. These results are reported in Figure 4 under the label "SWAN" as a representative proxy for stateless optimizers.

One technical consideration with these methods is their compatibility with gradient accumulation, which is a common practice in low-resource settings. In such scenarios, the fused backward strategy may not be applicable, and persistent gradient buffers are still required in memory (see Figure 11). While stateless optimizers are promising from a conceptual standpoint, this constraint currently poses practical challenges for achieving memory efficiency on par with parameter-efficient fine-tuning approaches like LoRA, or subspace-based approaches like fused GaLore and MoFaSGD.

## 6 Conclusion

We introduced MoFaSGD, a memory-efficient optimizer that uses memory comparable to LoRA-like methods while achieving strong convergence. The core idea is to maintain a low-rank factorization of momentum as the optimizer state and leverage this representation to directly update model parameters at each iteration using spectrally normalized updates, building upon the success of similar approaches in full-training scenarios. We provide a comprehensive theoretical analysis and establish an upper bound on convergence for stochastic non-convex optimization, matching existing lower bounds. Moreover, we empirically demonstrate the effectiveness of our method compared to standard low-rank optimization approaches in both pre-training and post-training setups.

**Limitations and Future Directions.** MoFaSGD shows strong empirical performance, but several limitations suggest promising directions for future work. First, as a low-rank subspace optimizer, MoFaSGD may underperform compared to full-rank methods on more complex tasks. Exploring adaptive or dynamic rank selection strategies, beyond the fixed-rank settings used in our study, could help mitigate this gap; for instance, future work could investigate monitoring projection residuals to allocate more rank to layers with higher approximation error or implementing a budget-aware allocation scheme to optimally balance performance and memory. Second, although our theoretical results establish convergence guarantees, they rely on assumptions such as nuclear-norm smoothness. The practical relevance and limitations of such assumptions in deep learning setups remain an open area for study.

## Acknowledgment

This work was partially supported by NSF CAREER Award #2239374.

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

## A   Additional Related Works

**Parameter-Efficient Fine-Tuning (PEFT).** To reduce the memory associated with full-parameter updates, PEFT methods constrain updates to a small subset of parameters or add minimal trainable components. Adapters introduce small bottleneck layers into the model (Houlsby et al., 2019), while the popular LoRA technique (Hu et al., 2021) injects trainable low-rank matrices into existing weight layers, drastically reducing memory by training only these adapters (Hu et al., 2021). Variants include AdaLoRA for adaptive rank allocation (Zhang et al., 2023), VeRA using shared low-rank matrices (Kopiczko et al., 2023), and DoRA decomposing updates into magnitude/direction (Liu et al., 2024). Other PEFT methods tune only biases (BitFit (Zaken et al., 2021)) or learn input embeddings (prompt tuning (Li & Liang, 2021; Lester et al., 2021)). QLoRA (Dettmers et al., 2024) further combines LoRA with 4-bit weight quantization. While memory-efficient, PEFT methods generally keep the base model fixed and rely on the capacity of the trained low-rank components.

**Second-Order and Preconditioning Methods.**

These methods aim primarily to accelerate convergence and improve optimization performance by incorporating curvature information, offering potentially more effective descent directions compared to first-order methods (George et al., 2018; Duvvuri et al., 2024; Gupta et al., 2018). Exact second-order information (e.g., using the full Hessian in Newton's method or the full gradient covariance in full-matrix Adagrad (Duchi et al., 2011; Duvvuri et al., 2024)) is computationally infeasible for large models due to the prohibitive cost ($\mathcal{O}(d^2)$ memory, $\mathcal{O}(d^3)$ computation) of storing and manipulating the required matrices (George et al., 2018; Duvvuri et al., 2024; Gupta et al., 2018). Therefore, practical methods rely on approximations to make harnessing second-order information tractable. Quasi-Newton methods like LBFGS (Liu & Nocedal, 1989; Duvvuri et al., 2024) build implicit Hessian approximations but can still be memory-intensive (Duvvuri et al., 2024).

A major family of approximations leverages Kronecker product factorizations. KFAC (Kronecker-Factored Approximate Curvature) (Martens & Grosse, 2015) approximates the Fisher Information Matrix (an approximation of the Hessian (Martens, 2020)) block-diagonally using Kronecker products specific to network layer types (Martens & Grosse, 2015; Grosse & Martens, 2016). Shampoo (Gupta et al., 2018), motivated by full-matrix Adagrad (Gupta et al., 2018; Anil et al., 2019), uses Kronecker products of gradient statistics ($\boldsymbol{G}\boldsymbol{G}^\top, \boldsymbol{G}^\top\boldsymbol{G}$) as a preconditioner (Gupta et al., 2018). While powerful, these factored approximations require computing matrix roots or inverses, which are computationally demanding and often amortized over multiple steps, potentially using stale curvature estimates (Gupta et al., 2018; George et al., 2018).

Refinements seek to improve accuracy or efficiency. EKFAC (George et al., 2018) builds on KFAC by performing cheaper diagonal updates within the Kronecker-factored eigenbasis (KFE), yielding a provably better Fisher approximation (George et al., 2018). CASPR (Duvvuri et al., 2024) uses Kronecker sums, aiming for a potentially better Adagrad approximation than Shampoo's Kronecker product (Duvvuri et al., 2024). SOAP (Vyas et al., 2024) runs AdamW within Shampoo's eigenbasis, aiming for improved stability and fewer hyperparameters, especially when the basis update is infrequent (Vyas et al., 2024). These methods highlight the ongoing effort to balance the power of second-order information with computational feasibility (George et al., 2018; Vyas et al., 2024).

**On-the-Fly and Partial Updates.** These techniques reduce memory by avoiding large buffers or distributing states. LOMO (Lv et al., 2023b) applies updates immediately, AdaLOMO (Lv et al., 2023a) adds adaptivity, BAdam (Luo et al., 2024) uses weight blocks, and ZeRO (Rajbhandari et al., 2020) shards states in distributed settings.

## B   Additional Preliminaries

**Adaptive Optimization Methods.** Adaptive optimization methods have become essential for training deep neural networks. By adjusting learning rates on a per-parameter (or per-dimension) basis using information from past gradients and the loss curvature, they often lead to significantly faster convergence than vanilla stochastic gradient descent (SGD) in practice. This line of work began with Adagrad (Duchi et al.,

2011), which proposed leveraging the gradient preconditioner $\boldsymbol{P}_t = \sum_{i=1}^{t} \boldsymbol{g}_i \boldsymbol{g}_i^\top \in \mathbb{R}^{mn \times mn}$ and performing the update rule as $\boldsymbol{w}_{t+1} = \boldsymbol{w}_t - \eta \boldsymbol{P}_t^{-\frac{1}{2}} \boldsymbol{g}_t$, where in our notation $\boldsymbol{g}_i = \text{Vec}(\boldsymbol{G}_i) \in \mathbb{R}^{mn}$ and $\boldsymbol{w}_t = \text{Vec}(\boldsymbol{W}_t)$. Note that considering only the diagonal terms of $\boldsymbol{P}_t$ and using the EMA of gradient covariances (e.g., $\boldsymbol{P}_t = \sum_{i=1}^{t} \beta_2^{t-i} \boldsymbol{g}_i \boldsymbol{g}_i^\top$) led to an early variant of RMSprop (Tieleman & Hinton, 2012), and further using EMA of the gradients themselves, $\hat{\boldsymbol{g}}_t = \sum_{i=1}^{t} \beta_1^{t-i} \boldsymbol{g}_i$, instead of the plain $\boldsymbol{g}_t$, yields the well-known Adam (Kingma & Ba, 2015). However, the mentioned methods are limited forms of second-order methods, where the preconditioner matrix is restricted to being diagonal. Considering either the original Adagrad (Duchi et al., 2011) perspective or second-order optimization methods such as Newton's method, moving beyond diagonal preconditioners is a natural step toward achieving even faster convergence. However, storing a non-diagonal preconditioner is often not feasible due to the size of DNNs, requiring $O(m^2 n^2)$ memory. Shampoo (Gupta et al., 2018) proposed maintaining a Kronecker approximation $\boldsymbol{P}_t \sim \boldsymbol{R}_t \otimes \boldsymbol{L}_t$, where $\boldsymbol{L}_t \in \mathbb{R}^{m \times m}$ and $\boldsymbol{R}_t \in \mathbb{R}^{n \times n}$, thereby reducing the number of parameters required for storing the preconditioning matrix to $O(m^2 + n^2)$. Morwani et al. (2024) showed that Shampoo (Gupta et al., 2018) closely approximates the full Adagrad preconditioner, unified all aforementioned methods, and argued that the gradient covariance $\boldsymbol{g}_t \boldsymbol{g}_t^\top$ closely approximates the Gauss-Newton components of the Hessian at $\boldsymbol{w}_t$, thereby drawing an interesting connection between Adagrad (Duchi et al., 2011) and second-order optimization methods.

## C Experimental Details

This section provides detailed configurations, hyperparameters, tuning procedures, and dataset information for the experiments presented in the main paper, aiming to ensure reproducibility.

### C.1 General Implementation Details

- **Software:** Experiments were implemented using standard libraries for deep learning, including PyTorch, Hugging Face Transformers, and Accelerate. Specific library versions are detailed in the code repository.

- **Hardware:** All experiments were conducted on NVIDIA A100 GPUs. The number of GPUs may have varied slightly depending on the specific experimental setup (e.g., 4 GPUs for Tulu3 tuning).

- **Code:** The implementation for MoFaSGD is available at `https://github.com/AnonCode1/MFSGD.git`.

### C.2 Pre-training: NanoGPT Speedrun

**Hyperparameters and Tuning.** For this benchmark, Muon hyperparameters were kept at the tuned defaults provided by Jordan et al. (2024a). Learning rates for AdamW, GaLore, and MoFaSGD were tuned via grid search over $\{1e-4, 2e-4, 3e-4, 5e-4, 8e-4, 1e-3, 3e-3, 5e-3, 8e-3, 1e-2, 2e-2, 5e-2\}$. MoFaSGD's momentum decay $\beta$ was tuned over $\{0.5, 0.85, 0.90, 0.95\}$. GaLore's SVD frequency was tuned over $\{10, 25, 75, 150, 300\}$. The best-performing hyperparameters based on final validation perplexity were selected.

Key hyperparameters selected for the NanoGPT pre-training experiments are summarized in Table 5.

### C.3 NLU Fine-tuning: GLUE Benchmark

**Hyperparameters and Tuning.** The experimental setup, including hyperparameters for baseline optimizers (AdamW, GaLore, LoRA+AdamW), directly follows the configuration reported in Zhao et al. (2024a) for RoBERTa-Base on GLUE. For our method, MoFaSGD, learning rates were tuned via grid search over $\{1e-5, 2e-5, 5e-5, 1e-4, 5e-4\}$ for each task and rank combination ($r = 4$ and $r = 8$), and the batch size was fixed to 16 for all experiments. The MoFaSGD momentum decay $\beta$ was also kept fixed at 0.95. The learning rate yielding the best validation accuracy on each specific task was selected. The final selected hyperparameters for MoFaSGD across the evaluated GLUE tasks are presented in Table 6.

Table 5: Final Selected Hyperparameters for NanoGPT Pre-training Experiment ($0.73B$ tokens)

| Optimizer | Rank | Learning Rate (LR) | SVD Freq. | MoFaSGD $\beta$ |
|---|---|---|---|---|
| Muon | - | 5e-2 | N/A | N/A |
| AdamW | - | 2e-3 | N/A | N/A |
| GaLore | $r = 16$ | 2e-2 | 150 | N/A |
| | $r = 32$ | 8e-3 | 75 | N/A |
| | $r = 128$ | 8e-3 | 75 | N/A |
| MoFaSGD (Ours) | $r = 16$ | 1e-3 | N/A | 0.85 |
| | $r = 32$ | 5e-4 | N/A | 0.85 |
| | $r = 128$ | 3e-4 | N/A | 0.85 |
| Batch Size (Tokens) | | 524,288 tokens | | |
| LR Schedule | | Stable then linear decay with cool-down of 0.4 | | |

Table 6: Final Selected MoFaSGD Hyperparameters for GLUE Tasks (RoBERTa-Base).

| Hyperparameter | Rank | MNLI | QQP | SST-2 | MRPC | COLA | QNLI | RTE |
|---|---|---|---|---|---|---|---|---|
| Learning Rate (LR) | $r = 4$ | 1e-4 | 5e-5 | 5e-5 | 1e-4 | 2e-5 | 2e-5 | 5e-5 |
| | $r = 8$ | 5e-5 | 5e-5 | 2e-5 | 5e-5 | 1e-5 | 1e-5 | 5e-5 |
| MoFaSGD $\beta$ | $r = 4, \ 8$ | | | | 0.95 | | | |
| Batch Size | $r = 4, \ 8$ | | | | 16 | | | |
| Epochs | $r = 4, \ 8$ | | | | 15 | | | |

## C.4 Instruction Tuning: Tulu3

**Hyperparameters and Tuning.** The setup largely follows Lambert et al. (2024). Learning rates for all optimizers were selected from the grid $\{1e-5, 5e-5, 1e-4, 5e-4, 1e-3\}$. The final LR for each method was chosen based on final validation loss on a held-out subset of the tulu-3-sft mixture. We used 5% of the sampled dataset for validation. Final selected hyperparameters for the Tulu3 instruction tuning experiment are summarized in Table 7.

Table 7: Final Selected Hyperparameters for Tulu-3 Instruction Tuning

| Hyperparameter | GaLore | LoRA | MoFaSGD (Ours) |
|---|---|---|---|
| Learning Rate (LR) | 5e-5 | 5e-5 | 1e-4 |
| Rank ($r$) | 8 | 8 | 8 |
| LoRA Alpha | N/A | 16 | N/A |
| GaLore SVD Freq. | 200 | N/A | N/A |
| MoFaSGD $\beta$ | N/A | N/A | 0.95 |
| Effective Batch Size | | 128 | |
| Epochs | | 1 | |

**Momentum Spectral Analysis.**

- **Methodology:** Analysis was performed on the Tulu3 instruction-tuning run using the default optimizer AdamW with a learning rate of $1e-5$ and a batch size of 128 for one epoch. We directly inspected the AdamW state buffer for the first-moment EMA ($M_t$) (state['exp_avg']) after each

backward pass periodically. SVD was performed on these buffers, and the energy ratio for rank $r$ was computed as $\frac{\sum_{i=1}^{r} \sigma_{i,M_t}^2}{\|M_t\|_F^2}$.

## C.5 Training Loss Curves

The training loss curves in Figures 8a and 8b illustrate MoFaSGD's convergence behavior. In both the simpler GLUE fine-tuning and the complex Tulu3 instruction-tuning, MoFaSGD consistently achieves lower training loss compared to LoRA and GaLore. This suggests more effective optimization dynamics, potentially stemming from the synergistic effect of adaptive low-rank momentum factorization and the spectrally normalized updates, which might offer better preconditioning than the standard updates used in LoRA or the subspace accumulation in GaLore.

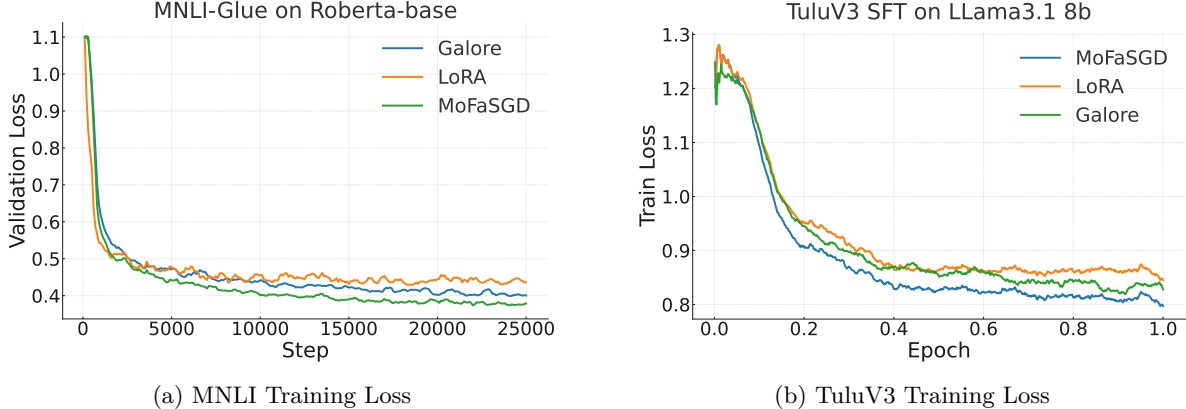

(a) MNLI Training Loss

(b) TuluV3 Training Loss

Figure 8: Training loss curves for post-training setups.

## C.6 Memory Profiling Details

We provide full memory trace visualizations and a quantitative breakdown of memory usage across all optimizer configurations (LLaMA3.1-8B, BF16, no activation checkpointing, batch size 1, gradient accumulation 8). These results correspond to the experiments summarized in Figure 4 and Section 5.4.

**Profiling Setup.** Memory traces were collected using PyTorch's native CUDA memory snapshot utility `torch.cuda.memory_snapshot()` at each training step. Each trace illustrates the evolution of GPU memory usage over four training steps. Manual annotations highlight key memory components.

**Observations.** As shown in the traces, MoFaSGD maintains a compact and stable memory profile, with minimal transient gradients and compressed optimizer state bands. AdamW and GaLore (non-fused) show persistent high memory usage due to full-rank gradient and optimizer state buffers. LoRA adds moderate adapter overhead, while fused GaLore shows improvements over its non-fused version.

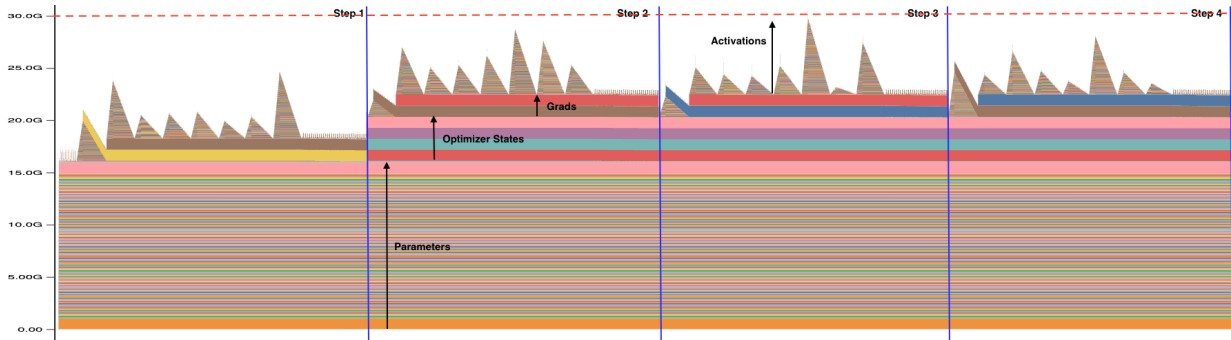

Figure 9: MoFaSGD ($r = 8$): Compact optimizer states and minimal gradient spikes. Total memory ∼29.4 GB.

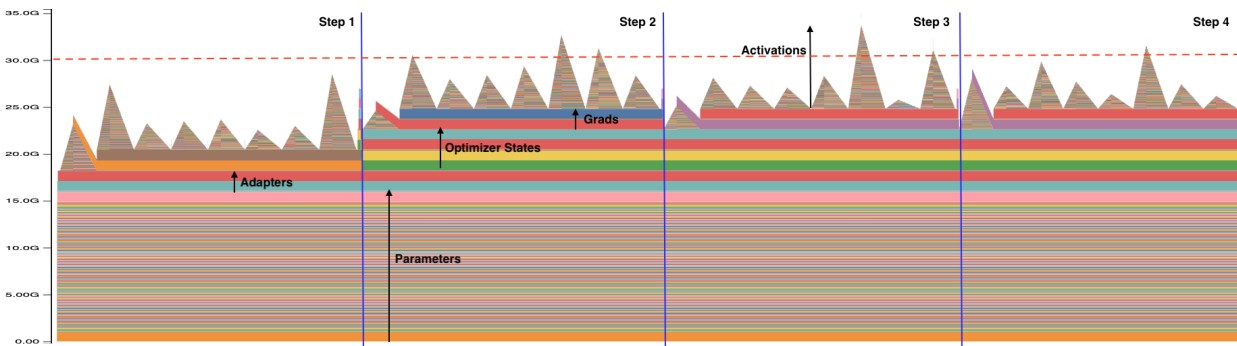

Figure 10: LoRA ($r = 8$): Slightly higher activation and adapter memory, total ∼33.6 GB.

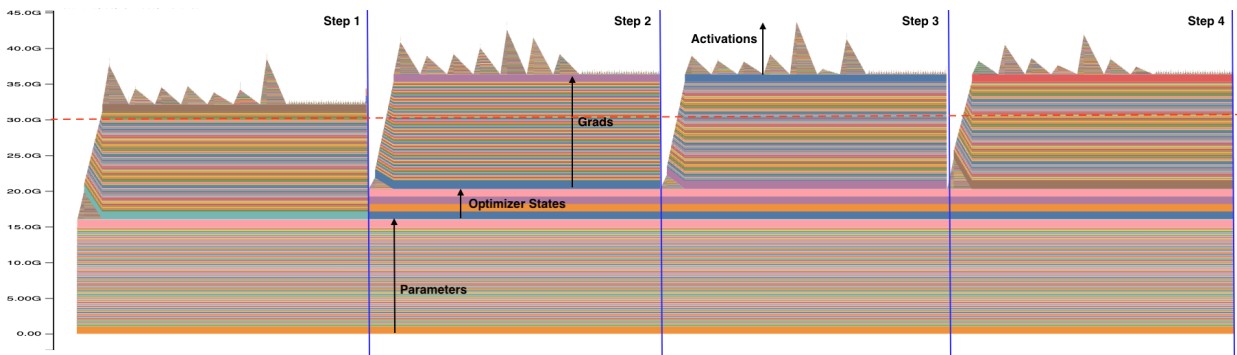

Figure 11: SWAN: Stateless optimizer, but full-sized gradient buffers lead to higher memory usage (∼43.9 GB).

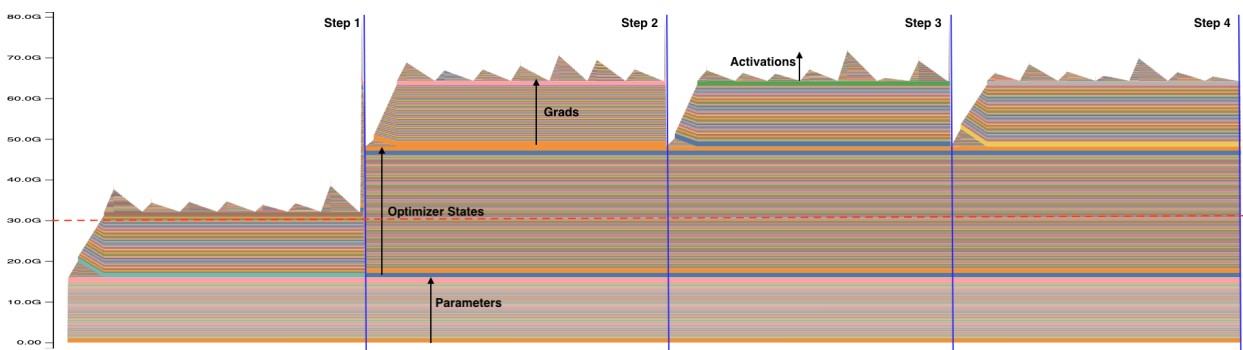

Figure 12: AdamW (BF16): Full-rank moments and gradient buffers dominate memory (∼70.8 GB).

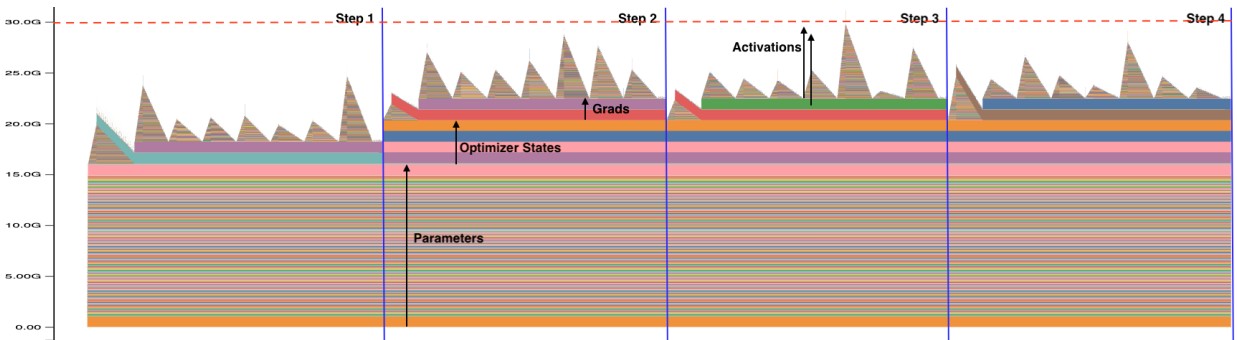

Figure 13: GaLore (Fused, $r = 8$): Low gradient and optimizer state memory; total ∼30.0 GB.

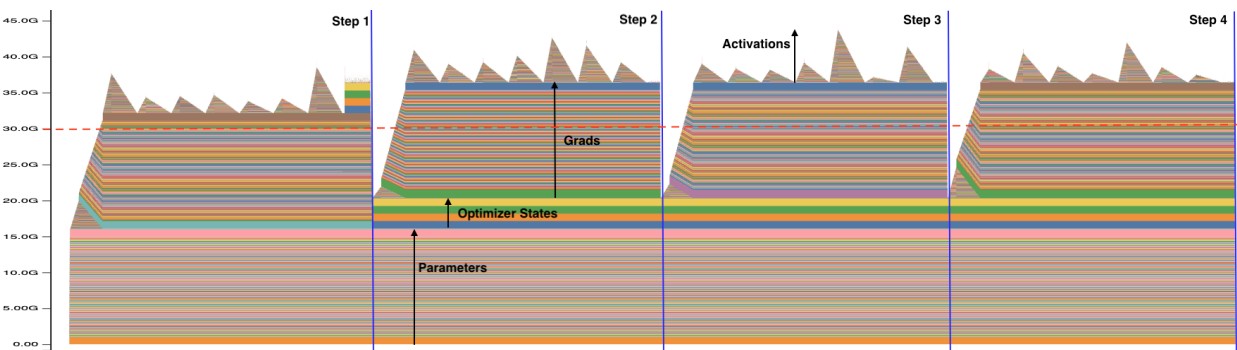

Figure 14: GaLore (Non-Fused, $r = 8$): Gradient accumulation inflates memory cost (∼44.5 GB).

**Quantitative Breakdown.** The table below summarizes the memory footprint by category for all setups.

Table 8: Memory breakdown (in GB) for LLaMA3.1-8B fine-tuning.

| Optimizer | Params | Opt. States | Gradients | Activations | Adapters |
|---|---|---|---|---|---|
| MoFaSGD ($r = 8$) | 15.5 | 4.2 | 2.1 | 7.6 | 0 |
| LoRA ($r = 8$) | 15.5 | 4.2 | 2.1 | 9.8 | 2.0 |
| SWAN | 15.5 | 4.2 | 16.0 | 8.2 | 0 |
| AdamW (BF16) | 15.5 | 31.8 | 16.0 | 7.5 | 0 |
| GaLore Fused ($r = 8$) | 15.5 | 4.2 | 2.1 | 8.2 | 0 |
| GaLore Non-Fused ($r = 8$) | 15.5 | 4.2 | 16.0 | 8.8 | 0 |

# D  Proofs and Technical Details

## D.1  On the Choice of Nuclear-norm Smoothness

One of the main assumptions behind our convergence bound in Theorem 4.5 is smoothness under the nuclear norm (Assumption 4.1). Here, we provide a detailed argument to justify this assumption. It is primarily motivated by concepts of descent in normed spaces, duality maps, and modular duality in deep learning—advancements detailed in recent works (Bernstein et al., 2023; Large et al., 2025; Bernstein & Newhouse, 2024a;b). These provide a solid foundation for spectral normalization of the descent update direction, which has been popularized in strong optimizers such as Muon (Jordan et al., 2024b; Liu et al., 2025).

First, we briefly discuss modular optimization theory, inspired by Large et al. (2025). The core idea of gradient descent is to optimize a locally linear upper bound of the loss function, as characterized by the gradient. Given a loss function $\mathcal{L} : \mathcal{W} \to \mathbb{R}$, we ideally want to find a tight upper bound on the higher-order terms, $\mathcal{R}(\boldsymbol{w}, \Delta\boldsymbol{w})$:

$$\mathcal{L}(\boldsymbol{w} + \Delta\boldsymbol{w}) \leq \mathcal{L}(\boldsymbol{w}) + \langle \nabla\mathcal{L}(\boldsymbol{w}), \Delta\boldsymbol{w} \rangle + \mathcal{R}(\boldsymbol{w}, \Delta\boldsymbol{w}) \tag{13}$$

For $\mathcal{W} \in \mathbb{R}^d$, classical second-order theory estimates the upper bound on higher-order terms using the maximum singular value of the Hessian. This leads to employing the $\ell_2$ norm with the typical upper bound $\frac{L}{2}\|\Delta\boldsymbol{w}\|_2^2$, where $L$ is the smoothness constant. However, this assumption often does not hold in deep learning practice and may yield a very loose upper bound. Modular theory (Large et al., 2025) aims to go beyond this by introducing an architecture-aware version of the upper bound based on the following conjecture:

$$\mathcal{L}(\boldsymbol{w} + \Delta\boldsymbol{w}) \leq \mathcal{L}(\boldsymbol{w}) + \langle \nabla\mathcal{L}(w), \Delta\boldsymbol{w} \rangle + \frac{\lambda}{2}\|\Delta\boldsymbol{w}\|^2 \qquad \text{(Normed Space Steepest Descent)}$$

Here, $\mathcal{W}$ can be any vector space, and $\|.\| : \mathcal{W} \to \mathbb{R}$ is an arbitrary norm on it. Modular optimization theory aims to find appropriate norms on the weight space of all model parameters $\mathcal{W} = \mathcal{W}_1 \times \cdots \times \mathcal{W}_L$. The modular norm on this parameter product space $\mathcal{W}$ is defined as the scaled max over individual norms (e.g., $\max(s_1\|\boldsymbol{w}_1\|_{\mathcal{W}_1}, \ldots, \|\boldsymbol{w}_L\|_{\mathcal{W}_L})$). The specific choice of norms for each parameter is based on its role and properties. For instance, for a linear layer $\boldsymbol{W}$, we expect the normalized feature change not to be drastic, which is characterized by the induced operator norm defined as $\|\Delta\boldsymbol{W}\|_{\alpha\to\beta} = \max \frac{\|\boldsymbol{W}\boldsymbol{x}\|_\beta}{\|\boldsymbol{x}\|_\alpha}$. Empirical observations show that for typical gradient updates, the normalized feature change of a linear layer is indeed close to the RMS-RMS operator norm (Large et al., 2025).

Large et al. (2025) show that by assigning norms to atomic modules (e.g., RMS $\to$ RMS for Linear layers and $\ell_1 \to$ RMS for embeddings), the modular norm of general architectures like Transformers satisfies the smoothness property. Informally, Proposition 5 of Large et al. (2025) shows that for a general module $\boldsymbol{M}$ over $\mathcal{X}, \mathcal{Y}, \mathcal{W}$, and for common loss functions $\mathcal{L}(\boldsymbol{w}) = \mathbb{E}_{x,y}[\ell(M(\boldsymbol{w}, \boldsymbol{x}))]$ such as cross-entropy, we have:

$$\|\nabla\mathcal{L}(\boldsymbol{w} + \Delta\boldsymbol{w}) - \nabla\mathcal{L}(\boldsymbol{w})\|_M^* \leq L_M\|\Delta\boldsymbol{w}\|_M \tag{14}$$

where $\|.\|_M^*$ is the dual norm of the modular norm defined earlier. Since our method targets the linear layers of Transformers, and the dual norm of RMS $\to$ RMS is the fan-in/fan-out scaled nuclear norm, we argue that assuming smoothness under the nuclear norm is more natural than under the Frobenius norm. The Frobenius norm assumption essentially treats the parameter space as $\mathbb{R}^d$, ignoring the matrix structure of linear layers and their architectural role.

However, note that our Assumption 4.1 is still stronger than Equation 14. Even if we use the RMS $\to$ RMS operator norm for all linear layers and assume the remaining parameters are fixed, letting the weight space be $\mathcal{W}_{\texttt{linear}} = (\boldsymbol{W}_1, \ldots, \boldsymbol{W}_{N_l})$, then by Equation 14, we would have:

$$\sum_{i=1}^{N_l} \|\nabla\mathcal{L}(\boldsymbol{W}_i + \Delta\boldsymbol{W}_i) - \nabla\mathcal{L}(\boldsymbol{W}_i)\|_* \leq L \max_{i=1}^{N_l} \left(\|\Delta\boldsymbol{W}_i\|_2\right),$$

where $\|.\|_*$ denotes the nuclear norm, and $\|.\|_2$ is the spectral norm, i.e., the $\ell_2 \to \ell_2$ operator norm.

## D.2 Mathematical Tools and Notations

**Kronecker Product and Vectorization.** Here, we introduce the background regarding the definition of Kronecker product, Vectorization, and well-known properties of these operators that make it easier to work with them. The Kronecker product is denoted as $\otimes$, and for any arbitrary $\boldsymbol{X} \in \mathbb{R}^{m_1 \times n_1}$ and $\boldsymbol{Y} \in \mathbb{R}^{m_2 \times n_2}$, is defined as follows, where $x_{i,j}$ are the elements on row $i$ and column $j$ of the matrix $\boldsymbol{X}$:

$$\boldsymbol{X} \otimes \boldsymbol{Y} = \begin{bmatrix} x_{1,1}\boldsymbol{Y} & x_{1,2}\boldsymbol{Y} & \cdots & x_{1,n_1}\boldsymbol{Y} \\ x_{2,1}\boldsymbol{Y} & x_{2,2}\boldsymbol{Y} & \cdots & x_{2,n_1}\boldsymbol{Y} \\ \vdots & \vdots & \ddots & \vdots \\ x_{m_1,1}\boldsymbol{Y} & x_{m_1,2}\boldsymbol{Y} & \cdots & x_{m_1,n_1}\boldsymbol{Y} \end{bmatrix} \tag{15}$$

Moreover, we define the vectorization operator Vec(.) that stacks columns of the matrix as a vector. Particularly, we have:

$$\mathrm{Vec}(\boldsymbol{X}) = \begin{bmatrix} x_{1,1} \\ x_{2,1} \\ \vdots \\ x_{m_1,1} \\ x_{1,2} \\ x_{2,2} \\ \vdots \\ x_{m_1,n_1} \end{bmatrix} \in \mathbb{R}^{m_1 n_1} \tag{16}$$

The following lemma summarizes basic properties of $\otimes$ and Vec(.) that we leverage throughout our proofs in the paper. Since this lemma covers basic, well-known properties in Matrix Algebra, we refer the reader to Horn & Johnson (1994) for proofs and details.

**Lemma D.1.** *Let A, B, C and D be arbitrary matrices and assume that operations in each property are well-defined with respect to their dimensions. We have:*

1. *Scalar Multiplication:* $\boldsymbol{A} \otimes (\alpha\boldsymbol{B}) = \alpha(\boldsymbol{A} \otimes \boldsymbol{B}) = (\alpha\boldsymbol{A}) \otimes \boldsymbol{B}$

2. *Associativity:* $(\boldsymbol{A} \otimes \boldsymbol{B}) \otimes \boldsymbol{C} = \boldsymbol{A} \otimes (\boldsymbol{B} \otimes \boldsymbol{C})$

3. *Distributivity over addition:* $(\boldsymbol{A}+\boldsymbol{B})\otimes\boldsymbol{C} = (\boldsymbol{A}\otimes\boldsymbol{C})+(\boldsymbol{B}\otimes\boldsymbol{C})$, *and* $\boldsymbol{A}\otimes(\boldsymbol{B}+\boldsymbol{C}) = (\boldsymbol{A}\otimes\boldsymbol{B})+(\boldsymbol{A}\otimes\boldsymbol{C})$

4. *Not necessarily commutative:* $(\boldsymbol{A} \otimes \boldsymbol{B}) \neq (\boldsymbol{B} \otimes \boldsymbol{A})$

5. *Transpose:* $(\boldsymbol{A} \otimes \boldsymbol{B})^\top = \boldsymbol{A}^\top \otimes \boldsymbol{B}^\top$

6. if $\boldsymbol{A}$ and $\boldsymbol{B}$ are positive-semidefinite, we have $(\boldsymbol{A} \otimes \boldsymbol{B})^s = \boldsymbol{A}^s \otimes \boldsymbol{B}^s$ for any positive real $s$, and if $\boldsymbol{A}$ is positive definite, it holds for any real $s$.

7. Mixed-product property: $(\boldsymbol{A} \otimes \boldsymbol{B})(\boldsymbol{C} \otimes \boldsymbol{D}) = \boldsymbol{AC} \otimes \boldsymbol{BD}$

8. Outer product: $\boldsymbol{u} \otimes \boldsymbol{v} = \boldsymbol{uv}^\top$

9. Mixed Kronecker matrix-vector product: $(\boldsymbol{A}^\top \otimes \boldsymbol{B}) \operatorname{Vec}(\boldsymbol{C}) = \operatorname{Vec}(\boldsymbol{BCA})$

### D.3 Proofs

*Proof of Theorem 4.3.* To prove Theorem 4.3, we first introduce the following lemma.

**Lemma D.2.** *Consider the reduced SVD decomposition of the left and right sketches as follows $\boldsymbol{L} = \boldsymbol{U_L \Sigma_L V_L}^\top$ and $\boldsymbol{R} = \boldsymbol{U_R \Sigma_R V_R}^\top$. Then, the vectorized unsketched representation of any arbitrary matrix $\boldsymbol{G} \in \mathbb{R}^{m \times n}$ can be decomposed as follows:*

$$\operatorname{Vec}(\boldsymbol{LL}^\top \boldsymbol{G} + \boldsymbol{GRR}^\top) = \boldsymbol{U_{L,R} \Sigma_{L,R} U_{L,R}^\top} \operatorname{Vec}(\boldsymbol{G}) \tag{17}$$

*where $\boldsymbol{U_{L,R}} \in \mathbb{R}^{mn \times r(m+n-r)}$ is a semi-orthogonal matrix (i.e., $\boldsymbol{U_{L,R}^\top U_{L,R}} = \boldsymbol{I}_{r(m+n-r)}$), and $\boldsymbol{\Sigma_{L,R}} \in \mathbb{R}^{(m+n-r) \times (m+n-r)}$ is a diagonal matrix, defined as below:*

$$\boldsymbol{U_{L,R}} = \begin{bmatrix} \boldsymbol{U_R} \otimes \boldsymbol{U_L}, \boldsymbol{U_R} \otimes \boldsymbol{U_L^\perp}, \boldsymbol{U_R^\perp} \otimes \boldsymbol{U_L} \end{bmatrix}$$
$$\boldsymbol{\Sigma_{L,R}} = \begin{bmatrix} \boldsymbol{\Sigma_R^2} \otimes \boldsymbol{I}_r + \boldsymbol{I}_r \otimes \boldsymbol{\Sigma_L^2} & \boldsymbol{0} & \boldsymbol{0} \\ \boldsymbol{0} & \boldsymbol{\Sigma_R^2} \otimes \boldsymbol{I}_{n-r} & \boldsymbol{0} \\ \boldsymbol{0} & \boldsymbol{0} & \boldsymbol{I}_{m-r} \otimes \boldsymbol{\Sigma_L^2} \end{bmatrix} \tag{18}$$

*Proof of Lemma D.2.* Recall the left singular vectors of the left and right sketch matrices, $\boldsymbol{U_L} \in \mathbb{R}^{m \times r}$ and $\boldsymbol{U_R} \in \mathbb{R}^{n \times r}$. We can augment these reduced bases with $m - r$ and $n - r$ orthogonal vectors to get full orthonormal bases as follows: $[\boldsymbol{U_L}, \boldsymbol{U_L^\perp}] \in \mathbb{R}^{m \times m}$ and $[\boldsymbol{U_R}, \boldsymbol{U_R^\perp}] \in \mathbb{R}^{n \times n}$. Since each of these expanded bases is square orthonormal matrices, we can write $\boldsymbol{U_L^\top U_L} = \boldsymbol{I}_r$, $(\boldsymbol{U_L^\perp})^\top \boldsymbol{U_L^\perp} = \boldsymbol{I}_{m-r}$, and also $\boldsymbol{U_L^\top U_L^\perp} = \boldsymbol{0}$. Similar properties hold for $\boldsymbol{U_R}$. Moreover, again based on the orthonormality of the augmented bases, we can write:

$$\boldsymbol{U_L U_L^\top} + \boldsymbol{U_L^\perp}(\boldsymbol{U_L^\perp})^\top = \boldsymbol{I}_m \quad , \quad \boldsymbol{U_R U_R^\top} + \boldsymbol{U_R^\perp}(\boldsymbol{U_R^\perp})^\top = \boldsymbol{I}_n \tag{19}$$

Leveraging Lemma D.1 we can write:

$$\begin{aligned} \operatorname{Vec}(\boldsymbol{LL}^\top \boldsymbol{G} + \boldsymbol{GRR}^\top) &= \operatorname{Vec}(\boldsymbol{LL}^\top \boldsymbol{G}) + \operatorname{Vec}(\boldsymbol{GRR}^\top) \\ &= \left(\boldsymbol{I}_n \otimes \boldsymbol{LL}^\top + \boldsymbol{RR}^\top \otimes \boldsymbol{I}_m\right) \operatorname{Vec}(\boldsymbol{G}) \\ &= \left(\boldsymbol{I}_n \otimes \boldsymbol{U_L \Sigma_L^2 U_L^\top} + \boldsymbol{U_R \Sigma_R^2 U_R^\top} \otimes \boldsymbol{I}_m\right) \operatorname{Vec}(\boldsymbol{G}) \end{aligned} \tag{20}$$

Using Equation 20 and properties in Lemma D.1, we can further write:

$$\begin{aligned} \boldsymbol{I}_n \otimes \boldsymbol{U_L \Sigma_L^2 U_L^\top} &= \left(\boldsymbol{U_R U_R^\top} + \boldsymbol{U_R^\perp}(\boldsymbol{U_R^\perp})^\top\right) \otimes \left(\boldsymbol{U_L \Sigma_L^2 U_L^\top}\right) \\ &= \boldsymbol{U_R U_R^\top} \otimes \boldsymbol{U_L \Sigma_L^2 U_L^\top} + \boldsymbol{U_R^\perp}(\boldsymbol{U_R^\perp})^\top \otimes \boldsymbol{U_L \Sigma_L^2 U_L^\top} \\ &= \left(\boldsymbol{U_R} \otimes \boldsymbol{U_L \Sigma_L}\right)\left(\boldsymbol{U_R} \otimes \boldsymbol{U_L \Sigma_L}\right)^\top + \left(\boldsymbol{U_R^\perp} \otimes \boldsymbol{U_L \Sigma_L}\right)\left(\boldsymbol{U_R^\perp} \otimes \boldsymbol{U_L \Sigma_L}\right)^\top \\ &= \left(\boldsymbol{U_R} \otimes \boldsymbol{U_L}\right)\left(\boldsymbol{I}_r \otimes \boldsymbol{\Sigma_L^2}\right)\left(\boldsymbol{U_R} \otimes \boldsymbol{U_L}\right)^\top + \left(\boldsymbol{U_R^\perp} \otimes \boldsymbol{U_L}\right)\left(\boldsymbol{I}_{n-r} \otimes \boldsymbol{\Sigma_L^2}\right)\left(\boldsymbol{U_R^\perp} \otimes \boldsymbol{U_L}\right)^\top \end{aligned} \tag{21}$$

Using a similar derivation as in Equation 21, we obtain:

$$\boldsymbol{U_R \Sigma_R^2 U_R^\top} \otimes \boldsymbol{I}_m = \left(\boldsymbol{U_R} \otimes \boldsymbol{U_L}\right)\left(\boldsymbol{\Sigma_R^2} \otimes \boldsymbol{I}_r\right)\left(\boldsymbol{U_R} \otimes \boldsymbol{U_L}\right)^\top + \left(\boldsymbol{U_R} \otimes \boldsymbol{U_L^\perp}\right)\left(\boldsymbol{\Sigma_R^2} \otimes \boldsymbol{I}_{m-r}\right)\left(\boldsymbol{U_R} \otimes \boldsymbol{U_L^\perp}\right)^\top \tag{22}$$

Plugging Equation 21 and Equation 22 back into Equation 20, we get:

$$
\begin{aligned}
\operatorname{Vec}(\boldsymbol{LL}^\top\boldsymbol{G} + \boldsymbol{GRR}^\top) &= \Big[\big(\boldsymbol{U_R}\otimes\boldsymbol{U_L}\big)\big(\boldsymbol{I}_r\otimes\boldsymbol{\Sigma}_L^2\big)\big(\boldsymbol{U_R}\otimes\boldsymbol{U_L}\big)^\top + \big(\boldsymbol{U_R^\perp}\otimes\boldsymbol{U_L}\big)\big(\boldsymbol{I}_{n-r}\otimes\boldsymbol{\Sigma}_L^2\big)\big(\boldsymbol{U_R^\perp}\otimes\boldsymbol{U_L}\big)^\top \\
&\quad + \big(\boldsymbol{U_R}\otimes\boldsymbol{U_L}\big)\big(\boldsymbol{\Sigma}_R^2\otimes\boldsymbol{I}_r\big)\big(\boldsymbol{U_R}\otimes\boldsymbol{U_L}\big)^\top \\
&\quad + \big(\boldsymbol{U_R}\otimes\boldsymbol{U_L^\perp}\big)\big(\boldsymbol{\Sigma}_R^2\otimes\boldsymbol{I}_{m-r}\big)\big(\boldsymbol{U_R}\otimes\boldsymbol{U_L^\perp}\big)^\top\Big]\operatorname{Vec}(\boldsymbol{G}) \\
&= \Big[\big(\boldsymbol{U_R}\otimes\boldsymbol{U_L}\big)\big(\boldsymbol{I}_r\otimes\boldsymbol{\Sigma}_L^2 + \boldsymbol{\Sigma}_R^2\otimes\boldsymbol{I}_r\big)\big(\boldsymbol{U_R}\otimes\boldsymbol{U_L}\big)^T \\
&\quad + \big(\boldsymbol{U_R^\perp}\otimes\boldsymbol{U_L}\big)\big(\boldsymbol{I}_{n-r}\otimes\boldsymbol{\Sigma}_L^2\big)\big(\boldsymbol{U_R^\perp}\otimes\boldsymbol{U_L}\big)^\top \\
&\quad + \big(\boldsymbol{U_R}\otimes\boldsymbol{U_L^\perp}\big)\big(\boldsymbol{\Sigma}_R^2\otimes\boldsymbol{I}_{m-r}\big)\big(\boldsymbol{U_R}\otimes\boldsymbol{U_L^\perp}\big)^\top\Big]\operatorname{Vec}(\boldsymbol{G})
\end{aligned}
\tag{23}
$$

To complete the proof, it remains to show that the block matrix

$$
\boldsymbol{U_{L,R}} = \big[\boldsymbol{U_R}\otimes\boldsymbol{U_L},\, \boldsymbol{U_R^\perp}\otimes\boldsymbol{U_L},\, \boldsymbol{U_R}\otimes\boldsymbol{U_L^\perp}\big] \in \mathbb{R}^{mn\times r(m+n-r)}
\tag{24}
$$

is semi-orthogonal, i.e. $\boldsymbol{U_{L,R}^\top}\boldsymbol{U_{L,R}} = \boldsymbol{I}_{r(m+n-r)}$.

First, note that pairs of sub-blocks are mutually orthogonal. For instance,

$$
\big(\boldsymbol{U_R}\otimes\boldsymbol{U_L}\big)^\top\big(\boldsymbol{U_R^\perp}\otimes\boldsymbol{U_L}\big) = \big(\boldsymbol{U_R^\top}\boldsymbol{U_R^\perp}\big)\otimes\big(\boldsymbol{U_L^\top}\boldsymbol{U_L}\big) = \boldsymbol{0}.
$$

A similar argument applies to all other sub-block pairs.

Moreover, each sub-block is orthonormal in its own right. For example,

$$
\big(\boldsymbol{U_R}\otimes\boldsymbol{U_L}\big)^\top\big(\boldsymbol{U_R}\otimes\boldsymbol{U_L}\big) = \big(\boldsymbol{U_R^\top}\boldsymbol{U_R}\big)\otimes\big(\boldsymbol{U_L^\top}\boldsymbol{U_L}\big) = \boldsymbol{I}_r\otimes\boldsymbol{I}_r = \boldsymbol{I}_{r^2}.
$$

Hence, $\boldsymbol{U_{L,R}}$ is a semi-orthogonal, and the result follows. $\qquad\square$

We turn to proving Theorem 4.3.

Let the reconstructed gradient after projection be $\tilde{\boldsymbol{G}} = \operatorname{Proj}_{(\boldsymbol{L},\boldsymbol{R})}(\boldsymbol{G}) = \alpha_1\boldsymbol{LL}^\top\boldsymbol{G} + \alpha_2\boldsymbol{GRR}^\top + \alpha_3\boldsymbol{LL}^\top\boldsymbol{GRR}^\top$. We can write:

$$
\operatorname{Vec}(\tilde{\boldsymbol{G}}) = \operatorname{Vec}(\alpha_1\boldsymbol{LL}^\top\boldsymbol{G} + \alpha_2\boldsymbol{GRR}^\top) + \alpha_3(\boldsymbol{RR}^\top\otimes\boldsymbol{LL}^\top)\operatorname{Vec}(\boldsymbol{G})
\tag{25}
$$

Replacing $\boldsymbol{L}$ and $\boldsymbol{R}$ with their corresponding reduced SVD decompositions, we have:

$$
(\boldsymbol{RR}^\top\otimes\boldsymbol{LL}^\top) = \boldsymbol{U_R}\boldsymbol{\Sigma}_R^2\boldsymbol{U_R^\top}\otimes\boldsymbol{U_L}\boldsymbol{\Sigma}_L^2\boldsymbol{U_L^\top} = (\boldsymbol{U_R}\otimes\boldsymbol{U_L})(\boldsymbol{\Sigma}_R^2\otimes\boldsymbol{\Sigma}_L^2)(\boldsymbol{U_R}\otimes\boldsymbol{U_L})^\top
\tag{26}
$$

Using Equation 23 from Lemma D.2 and Equation 26, we can rewrite:

$$
\begin{aligned}
\operatorname{Vec}(\tilde{\boldsymbol{G}}) &= \Big[\big(\boldsymbol{U_R}\otimes\boldsymbol{U_L}\big)\big(\alpha_1\boldsymbol{I}_r\otimes\boldsymbol{\Sigma}_L^2 + \alpha_2\boldsymbol{\Sigma}_R^2\otimes\boldsymbol{I}_r + \alpha_3\boldsymbol{\Sigma}_R^2\otimes\boldsymbol{\Sigma}_L^2\big)\big(\boldsymbol{U_R}\otimes\boldsymbol{U_L}\big)^\top \\
&\quad + \alpha_2\big(\boldsymbol{U_R^\perp}\otimes\boldsymbol{U_L}\big)\big(\boldsymbol{I}_{n-r}\otimes\boldsymbol{\Sigma}_L^2\big)\big(\boldsymbol{U_R^\perp}\otimes\boldsymbol{U_L}\big)^\top \\
&\quad + \alpha_1\big(\boldsymbol{U_R}\otimes\boldsymbol{U_L^\perp}\big)\big(\boldsymbol{\Sigma}_R^2\otimes\boldsymbol{I}_{m-r}\big)\big(\boldsymbol{U_R}\otimes\boldsymbol{U_L^\perp}\big)^\top\Big]\operatorname{Vec}(\boldsymbol{G}) \\
&\triangleq \boldsymbol{P_{L,R}}\operatorname{Vec}(\boldsymbol{G})
\end{aligned}
\tag{27}
$$

where in the last equality, we define $\boldsymbol{P_{L,R}}\in\mathbb{R}^{mn\times mn}$ as the corresponding projection matrix of $\operatorname{Proj}_{(\boldsymbol{L},\boldsymbol{R})}$.

We can use Equation 19 to write:

$$
\begin{aligned}
\boldsymbol{I}_{mn} = \boldsymbol{I}_n\otimes\boldsymbol{I}_m &= (\boldsymbol{U_R}\otimes\boldsymbol{U_L})(\boldsymbol{U_R}\otimes\boldsymbol{U_L})^\top + (\boldsymbol{U_R^\perp}\otimes\boldsymbol{U_L})(\boldsymbol{U_R^\perp}\otimes\boldsymbol{U_L})^\top + (\boldsymbol{U_R}\otimes\boldsymbol{U_L^\perp})(\boldsymbol{U_R}\otimes\boldsymbol{U_L^\perp})^\top \\
&\quad + (\boldsymbol{U_R^\perp}\otimes\boldsymbol{U_L^\perp})(\boldsymbol{U_R^\perp}\otimes\boldsymbol{U_L^\perp})^\top
\end{aligned}
\tag{28}
$$

We now propose our Kronecker decomposition of the subspace projection residual.

$$\mathrm{Vec}(\tilde{\boldsymbol{G}} - \boldsymbol{G}) = (\boldsymbol{I}_n \otimes \boldsymbol{I}_m - \boldsymbol{P}_{\boldsymbol{L},\boldsymbol{R}})\,\mathrm{Vec}(\boldsymbol{G}) =$$

$$\begin{aligned}
\Big[ &\big(\boldsymbol{U}_{\boldsymbol{R}} \otimes \boldsymbol{U}_{\boldsymbol{L}}\big)\big(\alpha_1 \boldsymbol{I}_r \otimes \boldsymbol{\Sigma}_{\boldsymbol{L}}^2 + \alpha_2 \boldsymbol{\Sigma}_{\boldsymbol{R}}^2 \otimes \boldsymbol{I}_r + \alpha_3 \boldsymbol{\Sigma}_{\boldsymbol{R}}^2 \otimes \boldsymbol{\Sigma}_{\boldsymbol{L}}^2 - \boldsymbol{I}_r \otimes \boldsymbol{I}_r\big)\big(\boldsymbol{U}_{\boldsymbol{R}} \otimes \boldsymbol{U}_{\boldsymbol{L}}\big)^\top \\
&+ \big(\boldsymbol{U}_{\boldsymbol{R}}^{\perp} \otimes \boldsymbol{U}_{\boldsymbol{L}}\big)\big(\alpha_2 \boldsymbol{I}_{n-r} \otimes \boldsymbol{\Sigma}_{\boldsymbol{L}}^2 - \boldsymbol{I}_{n-r} \otimes \boldsymbol{I}_r\big)\big(\boldsymbol{U}_{\boldsymbol{R}}^{\perp} \otimes \boldsymbol{U}_{\boldsymbol{L}}\big)^\top \\
&+ \big(\boldsymbol{U}_{\boldsymbol{R}} \otimes \boldsymbol{U}_{\boldsymbol{L}}^{\perp}\big)\big(\alpha_1 \boldsymbol{\Sigma}_{\boldsymbol{R}}^2 \otimes \boldsymbol{I}_{m-r} - \boldsymbol{I}_r \otimes \boldsymbol{I}_{m-r}\big)\big(\boldsymbol{U}_{\boldsymbol{R}} \otimes \boldsymbol{U}_{\boldsymbol{L}}^{\perp}\big)^\top \\
&+ \big(\boldsymbol{U}_{\boldsymbol{R}}^{\perp} \otimes \boldsymbol{U}_{\boldsymbol{L}}^{\perp}\big)\big(\boldsymbol{U}_{\boldsymbol{R}}^{\perp} \otimes \boldsymbol{U}_{\boldsymbol{L}}^{\perp}\big)^\top \Big]\,\mathrm{Vec}(\boldsymbol{G}).
\end{aligned} \tag{29}$$

For simplicity let $\boldsymbol{A}_1 = \big(\boldsymbol{U}_{\boldsymbol{R}} \otimes \boldsymbol{U}_{\boldsymbol{L}}\big)\big(\alpha_1 \boldsymbol{I}_r \otimes \boldsymbol{\Sigma}_{\boldsymbol{L}}^2 + \alpha_2 \boldsymbol{\Sigma}_{\boldsymbol{R}}^2 \otimes \boldsymbol{I}_r + \alpha_3 \boldsymbol{\Sigma}_{\boldsymbol{R}}^2 \otimes \boldsymbol{\Sigma}_{\boldsymbol{L}}^2 - \boldsymbol{I}_r \otimes \boldsymbol{I}_r\big)\big(\boldsymbol{U}_{\boldsymbol{R}} \otimes \boldsymbol{U}_{\boldsymbol{L}}\big)^\top$, $\boldsymbol{A}_2 = \big(\boldsymbol{U}_{\boldsymbol{R}}^{\perp} \otimes \boldsymbol{U}_{\boldsymbol{L}}\big)\big(\alpha_2 \boldsymbol{I}_{n-r} \otimes \boldsymbol{\Sigma}_{\boldsymbol{L}}^2 - \boldsymbol{I}_{n-r} \otimes \boldsymbol{I}_r\big)\big(\boldsymbol{U}_{\boldsymbol{R}}^{\perp} \otimes \boldsymbol{U}_{\boldsymbol{L}}\big)^\top$, $\boldsymbol{A}_3 = \big(\boldsymbol{U}_{\boldsymbol{R}} \otimes \boldsymbol{U}_{\boldsymbol{L}}^{\perp}\big)\big(\alpha_1 \boldsymbol{\Sigma}_{\boldsymbol{R}}^2 \otimes \boldsymbol{I}_{m-r} - \boldsymbol{I}_r \otimes \boldsymbol{I}_{m-r}\big)\big(\boldsymbol{U}_{\boldsymbol{R}} \otimes \boldsymbol{U}_{\boldsymbol{L}}^{\perp}\big)^\top$ and $\boldsymbol{A}_4 = \big(\boldsymbol{U}_{\boldsymbol{R}}^{\perp} \otimes \boldsymbol{U}_{\boldsymbol{L}}^{\perp}\big)\big(\boldsymbol{U}_{\boldsymbol{R}}^{\perp} \otimes \boldsymbol{U}_{\boldsymbol{L}}^{\perp}\big)^\top$. First note that $\boldsymbol{A}_1, \boldsymbol{A}_2, \boldsymbol{A}_3, \boldsymbol{A}_4 \in \mathbb{R}^{mn \times mn}$. For all distinct $(i,j) \in \{1,2,3,4\}$, their matrix product is zero, $\boldsymbol{A}_i^\top \boldsymbol{A}_j = \boldsymbol{0}$. Without loss of generality, we show it for $\boldsymbol{A}_2$, and $\boldsymbol{A}_3$ here.

$$\begin{aligned}
\boldsymbol{A}_2^\top \boldsymbol{A}_3 &= \big(\boldsymbol{U}_{\boldsymbol{R}}^{\perp} \otimes \boldsymbol{U}_{\boldsymbol{L}}\big)\big(\alpha_2 \boldsymbol{I}_{n-r} \otimes \boldsymbol{\Sigma}_{\boldsymbol{L}}^2 - \boldsymbol{I}_{n-r} \otimes \boldsymbol{I}_r\big)\big(\boldsymbol{U}_{\boldsymbol{R}}^{\perp} \otimes \boldsymbol{U}_{\boldsymbol{L}}\big)^\top \big(\boldsymbol{U}_{\boldsymbol{R}} \otimes \boldsymbol{U}_{\boldsymbol{L}}^{\perp}\big)\big(\alpha_3 \boldsymbol{\Sigma}_{\boldsymbol{R}}^2 \otimes \boldsymbol{I}_{m-r} - \boldsymbol{I}_r \otimes \boldsymbol{I}_{m-r}\big) \\
&\quad \big(\boldsymbol{U}_{\boldsymbol{R}} \otimes \boldsymbol{U}_{\boldsymbol{L}}^{\perp}\big)^\top \\
&= \big(\boldsymbol{U}_{\boldsymbol{R}}^{\perp} \otimes \boldsymbol{U}_{\boldsymbol{L}}\big)\big(\alpha_2 \boldsymbol{I}_{n-r} \otimes \boldsymbol{\Sigma}_{\boldsymbol{L}}^2 - \boldsymbol{I}_{n-r} \otimes \boldsymbol{I}_r\big)\big(\boldsymbol{U}_{\boldsymbol{R}}^{\perp\top} \boldsymbol{U}_{\boldsymbol{R}} \otimes \boldsymbol{U}_{\boldsymbol{L}}^\top \boldsymbol{U}_{\boldsymbol{L}}^{\perp}\big)^\top \big(\alpha_3 \boldsymbol{\Sigma}_{\boldsymbol{R}}^2 \otimes \boldsymbol{I}_{m-r} - \boldsymbol{I}_r \otimes \boldsymbol{I}_{m-r}\big) \\
&\quad \big(\boldsymbol{U}_{\boldsymbol{R}} \otimes \boldsymbol{U}_{\boldsymbol{L}}^{\perp}\big)^\top \\
&= \big(\boldsymbol{U}_{\boldsymbol{R}}^{\perp} \otimes \boldsymbol{U}_{\boldsymbol{L}}\big)\big(\alpha_2 \boldsymbol{I}_{n-r} \otimes \boldsymbol{\Sigma}_{\boldsymbol{L}}^2 - \boldsymbol{I}_{n-r} \otimes \boldsymbol{I}_r\big)\big(\boldsymbol{0}_{n-r,r} \otimes \boldsymbol{0}_{n-r,r}\big)^\top \big(\alpha_3 \boldsymbol{\Sigma}_{\boldsymbol{R}}^2 \otimes \boldsymbol{I}_{m-r} - \boldsymbol{I}_r \otimes \boldsymbol{I}_{m-r}\big) \\
&\quad \big(\boldsymbol{U}_{\boldsymbol{R}} \otimes \boldsymbol{U}_{\boldsymbol{L}}^{\perp}\big)^\top \\
&= \boldsymbol{0}_{mn}
\end{aligned} \tag{30}$$

Using Equation 29, and above fact yields:

$$\|\tilde{\boldsymbol{G}} - \boldsymbol{G}\|_F^2 = \|\mathrm{Vec}(\tilde{\boldsymbol{G}} - \boldsymbol{G})\|_2^2 = \|\boldsymbol{A}_1 \boldsymbol{g}\|_2^2 + \|\boldsymbol{A}_2 \boldsymbol{g}\|_2^2 + \|\boldsymbol{A}_3 \boldsymbol{g}\|_2^2 + \|\boldsymbol{A}_4 \boldsymbol{g}\|_2^2 \tag{31}$$

where $\|.\|_2$ here is $\ell_2$ norm of a vector, and $\boldsymbol{g} = \mathrm{Vec}(\boldsymbol{G})$. The equality holds because $(\boldsymbol{A}_i \boldsymbol{g})^\top (\boldsymbol{A}_j \boldsymbol{g}) = \boldsymbol{0}_{mn}$. Based on the above equation, we have the following lower bound on residual $\|\tilde{\boldsymbol{G}} - \boldsymbol{G}\|_F^2 \geq \|\boldsymbol{A}_4 \boldsymbol{g}\|_2^2$ for any given $\boldsymbol{L}$, and $\boldsymbol{R}$. The term $\boldsymbol{A}_4$ does not depend on the choice of $(\alpha_1, \alpha_2, \alpha_3)$, also $\boldsymbol{\Sigma}_{\boldsymbol{R}}$ and $\boldsymbol{\Sigma}_{\boldsymbol{L}}$. Thus, the lower bound on this residual would be tight if and only if we have $\|\boldsymbol{A}_1 \boldsymbol{g}\|_2 = \|\boldsymbol{A}_2 \boldsymbol{g}\|_2 = \|\boldsymbol{A}_3 \boldsymbol{g}\|_2 = 0$. By setting $\boldsymbol{\Sigma}_{\boldsymbol{L}} = \boldsymbol{I}_r$, $\boldsymbol{\Sigma}_{\boldsymbol{R}} = \boldsymbol{I}_r$, and then considering $(\alpha_1, \alpha_2, \alpha_3) = (1, 1, -1)$, we have $\boldsymbol{A}_1 = \boldsymbol{A}_2 = \boldsymbol{A}_3 = \boldsymbol{0}_{mn}$.

Thus, we have shown that under conditions in Theorem 4.3, the first three terms in Equation 31 will be exactly zero, leaving the following term as the projection residual:

$$\mathrm{Vec}(\tilde{\boldsymbol{G}}_{\mathrm{optimal}} - \boldsymbol{G}) = \big(\boldsymbol{U}_{\boldsymbol{R}}^{\perp} \otimes \boldsymbol{U}_{\boldsymbol{L}}^{\perp}\big)\big(\boldsymbol{U}_{\boldsymbol{R}}^{\perp} \otimes \boldsymbol{U}_{\boldsymbol{L}}^{\perp}\big)^\top \mathrm{Vec}(\boldsymbol{G}). \tag{32}$$

$\square$

### D.3.1 Convergence Proofs

In this section, we aim to find an upper bound on the iteration complexity of Algorithm 1 for finding an $\epsilon$-stationary point defined by the averaged nuclear norm of gradients. First, we recall the necessary assumptions and notations.

**Update Rule.** Let $\{\boldsymbol{W}_t\}_{t=1}^T$ be the iterates of Algorithm 1. Then we have the following equivalent update rule:

$$\boldsymbol{W}_{t+1} = \boldsymbol{W}_t - \eta \boldsymbol{U}_{t+1} \boldsymbol{V}_{t+1}^\top \quad, \quad \hat{\boldsymbol{M}}_t = \boldsymbol{U}_{t+1} \boldsymbol{\Sigma}_{t+1} \boldsymbol{V}_{t+1}^\top \quad, \quad \boldsymbol{M}_t = \sum_{i=0}^t \beta^{t-i} \boldsymbol{G}_t \tag{33}$$

where $\boldsymbol{M}_t$ represents the full-rank momentum, and $\boldsymbol{G}_t = \nabla\mathcal{L}(\boldsymbol{W}_t, \xi_t) \in \mathbb{R}^{m \times n}$ is the stochastic gradient at parameter $\boldsymbol{W}_t$. Moreover, let $\hat{\boldsymbol{M}}_t$ be the low-rank momentum factorization, where $\boldsymbol{U}_i \in \mathbb{R}^{m \times r}$, $\boldsymbol{V}_i \in \mathbb{R}^{n \times r}$, and $\boldsymbol{\Sigma}_i \in \mathbb{R}^{r \times r}$.

**Definitions.** We recall the necessary definitions. We let the optimization objective be represented as $\mathcal{L}(\boldsymbol{W}) = \mathbb{E}_{\xi}[\mathcal{L}(\boldsymbol{W}, \xi)]$, and moreover assume we have access to an unbiased, variance-bounded ($\sigma$) gradient oracle. For any optimization iterate $\boldsymbol{W}_i$, the full-batch gradient is denoted as $\bar{\boldsymbol{G}}_i = \nabla\mathcal{L}(\boldsymbol{W}_i)$, and the stochastic gradient is denoted as $\boldsymbol{G}_i = \nabla\mathcal{L}(\boldsymbol{W}_i, \xi_i)$. Based on assumptions on the gradient oracle, we have $\bar{\boldsymbol{G}}_i = \mathbb{E}[\boldsymbol{G}_i]$, and $\mathbb{E}[\|\boldsymbol{G}_i - \bar{\boldsymbol{G}}_i\|_*] \leq \sigma$. Moreover, we say a function $\mathcal{L}(.) : \mathbb{R}^{m \times n} \to r$ is $L$-smooth with respect to an arbitrary norm $\|\cdot\|$ if for any two parameters $\boldsymbol{W}_1, \boldsymbol{W}_2$, we have $\|\nabla\mathcal{L}(\boldsymbol{W}_1) - \nabla\mathcal{L}(\boldsymbol{W}_2)\|_* \leq L\|\boldsymbol{W}_1 - \boldsymbol{W}_2\|_2$.

Next, we propose our descent lemma.

**Lemma D.3** (Descent lemma). *Let $\{\boldsymbol{W}\}_{t=0}^T$ be the iterates of Algorithm 1, optimized under a loss function that satisfies Assumption 4.1. Then, we have:*

$$\mathcal{L}(\boldsymbol{W}_{t+1}) \leq \mathcal{L}(\boldsymbol{W}_t) - \eta\|\bar{\boldsymbol{G}}_t\|_* + 2\eta\|\hat{\boldsymbol{M}}_t - \bar{\boldsymbol{G}}_t\|_* + \frac{\eta^2 L}{2} \tag{34}$$

*Proof of Lemma D.3.* First note that $\mathcal{L}(.)$ is $L$-smooth with the nuclear norm. Therefore for any $\boldsymbol{W}_1, \boldsymbol{W}_2 \in \mathbb{R}^{m \times n}$ we have: $\|\nabla\mathcal{L}(\boldsymbol{W}_1) - \nabla\mathcal{L}(\boldsymbol{W}_2)\|_* \leq L\|\boldsymbol{W}_1 - \boldsymbol{W}_2\|_2$, where $\|.\|_2$ is the spectral norm. Due to this property, we can leverage Proposition 5 in Large et al. (2025) to argue that for any $\boldsymbol{W}_1, \boldsymbol{W}_2$ we have:

$$\mathcal{L}(\boldsymbol{W}_2) \leq \mathcal{L}(\boldsymbol{W}_1) + \langle\nabla\mathcal{L}(\boldsymbol{W}_1), \boldsymbol{W}_2 - \boldsymbol{W}_1\rangle_F + \frac{L}{2}\|\boldsymbol{W}_1 - \boldsymbol{W}_2\|_2 \tag{35}$$

Thus, considering the update rule as in Equation 33, for two consecutive iterates $\boldsymbol{W}_t$ and $\boldsymbol{W}_{t+1}$, we can write:

$$\mathcal{L}(\boldsymbol{W}_{t+1}) \leq \mathcal{L}(\boldsymbol{W}_t) - \eta\langle\bar{\boldsymbol{G}}_t, \boldsymbol{U}_{t+1}\boldsymbol{V}_{t+1}^\top\rangle + \frac{L\eta^2}{2}\|\boldsymbol{U}_{t+1}\boldsymbol{V}_{t+1}^\top\|_2 \tag{36}$$

Note that if we take $\|.\|_*$ as a function, its subgradient is well-known and can be derived as follows: the subgradient set at $\boldsymbol{X} = \boldsymbol{U}\boldsymbol{\Sigma}\boldsymbol{V}^\top$ is $\partial\|\boldsymbol{X}\|_* = \{\boldsymbol{U}\boldsymbol{V}^\top + \boldsymbol{H} : \boldsymbol{U}^\top\boldsymbol{H} = 0, \boldsymbol{V}^\top\boldsymbol{H} = 0, \|\boldsymbol{W}\|_2 \leq 1\}$. Since the nuclear norm is a convex function, for any $\boldsymbol{X} = \boldsymbol{U}\boldsymbol{\Sigma}\boldsymbol{V}^\top$ and $\boldsymbol{Y}$, we have:

$$\|\boldsymbol{Y}\|_* \geq \|\boldsymbol{X}\|_* + \langle\boldsymbol{U}\boldsymbol{V}^\top, \boldsymbol{Y} - \boldsymbol{X}\rangle \tag{37}$$

Replace $\boldsymbol{Y} = \hat{\boldsymbol{M}}_t - \bar{\boldsymbol{G}}_t$ and $\boldsymbol{X} = \hat{\boldsymbol{M}}_t = \boldsymbol{U}_{t+1}\boldsymbol{\Sigma}_{t+1}\boldsymbol{V}_{t+1}^\top$; then we can rewrite Equation 36 as:

$$\mathcal{L}(\boldsymbol{W}_{t+1}) \leq \mathcal{L}(\boldsymbol{W}_t) - \eta\|\hat{\boldsymbol{M}}_t\|_* + \eta\|\hat{\boldsymbol{M}}_t - \bar{\boldsymbol{G}}_t\|_* + \frac{\eta^2}{2}L \tag{38}$$

where we also used the fact that $\|\boldsymbol{U}\boldsymbol{V}^\top\|_2 = 1$. Using the triangle inequality, we have $\|\hat{\boldsymbol{M}}_t\|_* \geq \|\bar{\boldsymbol{G}}_t\|_* - \|\hat{\boldsymbol{M}}_t - \bar{\boldsymbol{G}}_t\|_*$, and plugging this into Equation 38 completes the proof. $\qquad\square$

As indicated in Lemma D.3, the term $\|\hat{\boldsymbol{M}}_t - \bar{\boldsymbol{G}}_t\|_*$ is the main challenge for deriving the upper bound. Note that we can write $\|\hat{\boldsymbol{M}}_t - \bar{\boldsymbol{G}}_t\|_* \leq \|\boldsymbol{M}_t - \bar{\boldsymbol{G}}_t\|_* + \|\hat{\boldsymbol{M}}_t - \boldsymbol{M}_t\|_*$. Thus, we aim to bound each decomposed term separately. Note that the first decomposed term $\|\boldsymbol{M}_t - \bar{\boldsymbol{G}}_t\|_*$ measures the nuclear norm difference between the full-batched gradient and the first momentum. Moreover, the second decomposed term $\|\hat{\boldsymbol{M}}_t - \boldsymbol{M}_t\|_*$ measures the low-rank compression error of the full-rank momentum.

**Lemma D.4.** *Under Assumptions 4.1 and 4.1 we have:*

$$\sum_{t=1}^T \mathbb{E}[\|\boldsymbol{M}_t - \bar{\boldsymbol{G}}_t\|_*] \leq \frac{\beta}{1-\beta}\sum_{t=0}^{T-1}\mathbb{E}[\|\bar{\boldsymbol{G}}_t\|_*] + \frac{T\sigma}{(1-\beta)\sqrt{B}} \tag{39}$$

*Proof of Lemma D.4.* Let $\boldsymbol{\Delta}_t = \boldsymbol{G}_t - \bar{\boldsymbol{G}}_t$. We can write $\|\boldsymbol{M}_t - \bar{\boldsymbol{G}}_t\|_* \leq \|\boldsymbol{M}_t - \boldsymbol{G}_t\|_* + \|\boldsymbol{\Delta}_t\|_* = \beta\|\boldsymbol{M}_{t-1}\|_* + \|\boldsymbol{\Delta}_t\|_*$. Moreover, we have $\|\boldsymbol{M}_t\|_* \leq \|\boldsymbol{G}_t\|_* + \beta\|\boldsymbol{M}_{t-1}\|_*$, which, upon unrolling, gives

$\|\boldsymbol{M}_t\|_* \le \sum_{i=0}^t \beta^{t-i}\|\boldsymbol{G}_i\|_*$. Thus, we can write $\|\boldsymbol{M}_t - \boldsymbol{G}_t\|_* \le \beta\sum_{i=0}^{t-1}\beta^{t-1-i}\|\boldsymbol{G}_i\|_*$. Finally, since $\sum_{i=0}^j \beta^i = \frac{1-\beta^{j+1}}{1-\beta} \le \frac{1}{1-\beta}$, we have:

$$\sum_{t=1}^T \|\boldsymbol{M}_t - \bar{\boldsymbol{G}}_t\|_* \le \frac{\beta}{1-\beta}\sum_{t=0}^{T-1}\|\bar{\boldsymbol{G}}_t\|_* + \frac{1}{1-\beta}\sum_{t=0}^T \|\boldsymbol{\Delta}_t\|_* \tag{40}$$

Taking the expectation and leveraging Assumption 4.1, we have:

$$\sum_{t=1}^T \mathbb{E}[\|\boldsymbol{M}_t - \bar{\boldsymbol{G}}_t\|_*] \le \frac{\beta}{1-\beta}\sum_{t=0}^{T-1}\mathbb{E}[\|\bar{\boldsymbol{G}}_t\|_*] + \frac{T\sigma}{(1-\beta)\sqrt{B}} \tag{41}$$

$\square$

We now bound the second term $\|\hat{\boldsymbol{M}}_t - \boldsymbol{M}_t\|_*$. We have the following lemma:

**Lemma D.5.** *Let $\|\hat{\boldsymbol{M}}_t - \boldsymbol{M}_t\|_*$ be the compression error of the low-rank momentum estimation at iteration $t$. Then under Assumption 4.1 and the condition $\mathrm{rank}(\boldsymbol{G}_0) \le r$, we have the following bound:*

$$\sum_{t=1}^T \mathbb{E}\big[\|\hat{\boldsymbol{M}}_t - \boldsymbol{M}_t\|_*\big] \le \frac{\eta L T}{1-\beta} + \frac{2\sigma T}{\sqrt{B}(1-\beta)} \tag{42}$$

*Proof of Lemma D.5.* First note that we have:

$$\|\hat{\boldsymbol{M}}_t - \boldsymbol{M}_t\|_* = \|\beta\boldsymbol{M}_{t-1} + \boldsymbol{G}_t - \hat{\boldsymbol{M}}_t\|_* \le \beta\|\hat{\boldsymbol{M}}_{t-1} - \boldsymbol{M}_{t-1}\|_* + \|\boldsymbol{G}_t + \beta\hat{\boldsymbol{M}}_{t-1} - \hat{\boldsymbol{M}}_t\|_* \tag{43}$$

Considering the momentum factor update rule in 1, we can replace $\hat{\boldsymbol{M}}_t$ with $\hat{\boldsymbol{G}}_t + \beta\hat{\boldsymbol{M}}_{t-1}$, where $\hat{\boldsymbol{G}}_t = \boldsymbol{U}_t\boldsymbol{U}_t^\top\boldsymbol{G}_t + \boldsymbol{G}_t\boldsymbol{V}_t\boldsymbol{V}_t^\top - \boldsymbol{U}_t\boldsymbol{U}_t^\top\boldsymbol{G}_t\boldsymbol{V}_t\boldsymbol{V}_t^\top$. Therefore, we can write:

$$\|\hat{\boldsymbol{M}}_t - \boldsymbol{M}_t\|_* \le \beta\|\hat{\boldsymbol{M}}_{t-1} - \boldsymbol{M}_{t-1}\|_* + \|(\boldsymbol{I} - \boldsymbol{U}_t\boldsymbol{U}_t^\top)\boldsymbol{G}_t(\boldsymbol{I} - \boldsymbol{V}_t\boldsymbol{V}_t^\top)\|_* \tag{44}$$

The term $\|(\boldsymbol{I} - \boldsymbol{U}_t\boldsymbol{U}_t^\top)\boldsymbol{G}_t(\boldsymbol{I} - \boldsymbol{V}_t\boldsymbol{V}_t^\top)\|_*$ represents the low-rank compression error of gradients happening in our Algorithm 1. Thus, we aim to bound this term. We have:

$$\begin{aligned}
\|(\boldsymbol{I} - \boldsymbol{U}_t\boldsymbol{U}_t^\top)\boldsymbol{G}_t(\boldsymbol{I} - \boldsymbol{V}_t\boldsymbol{V}_t^\top)\|_* &\le \|(\boldsymbol{I} - \boldsymbol{U}_t\boldsymbol{U}_t^\top)(\boldsymbol{G}_t - \boldsymbol{G}_{t-1})(\boldsymbol{I} - \boldsymbol{V}_t\boldsymbol{V}_t^\top)\|_* + \|(\boldsymbol{I} - \boldsymbol{U}_t\boldsymbol{U}_t^\top)\boldsymbol{G}_{t-1}(\boldsymbol{I} - \boldsymbol{V}_t\boldsymbol{V}_t^\top)\|_* \\
&\le \|(\boldsymbol{I} - \boldsymbol{U}_t\boldsymbol{U}_t^\top)(\bar{\boldsymbol{G}}_t - \bar{\boldsymbol{G}}_{t-1})(\boldsymbol{I} - \boldsymbol{V}_t\boldsymbol{V}_t^\top)\|_* + \|(\boldsymbol{I} - \boldsymbol{U}_t\boldsymbol{U}_t^\top)\boldsymbol{G}_{t-1}(\boldsymbol{I} - \boldsymbol{V}_t\boldsymbol{V}_t^\top)\|_* + \|\boldsymbol{\Delta}_t - \boldsymbol{\Delta}_{t-1}\|_* \\
&\le L\|\boldsymbol{W}_t - \boldsymbol{W}_{t-1}\|_2 + \|(\boldsymbol{I} - \boldsymbol{U}_t\boldsymbol{U}_t^\top)\boldsymbol{G}_{t-1}(\boldsymbol{I} - \boldsymbol{V}_t\boldsymbol{V}_t^\top)\|_* + \|\boldsymbol{\Delta}_t - \boldsymbol{\Delta}_{t-1}\|_* \\
&\le \eta L + \|(\boldsymbol{I} - \boldsymbol{U}_t\boldsymbol{U}_t^\top)\boldsymbol{G}_{t-1}(\boldsymbol{I} - \boldsymbol{V}_t\boldsymbol{V}_t^\top)\|_* + \|\boldsymbol{\Delta}_t - \boldsymbol{\Delta}_{t-1}\|_*
\end{aligned} \tag{45}$$

where in the last inequality, we used the smoothness property in Assumption 4.1, and also the fact that $\boldsymbol{W}_t - \boldsymbol{W}_{t-1} = -\eta\boldsymbol{U}_t\boldsymbol{V}_t^\top$.

Next, we bound the term $\|(\boldsymbol{I} - \boldsymbol{U}_t\boldsymbol{U}_t^\top)\boldsymbol{G}_{t-1}(\boldsymbol{I} - \boldsymbol{V}_t\boldsymbol{V}_t^\top)\|_*$. First note that based on Equation 7, we can see that $\mathrm{Range}([\boldsymbol{U}_{t-1} \quad \boldsymbol{G}_{t-1}\boldsymbol{V}_{t-1}]) \subseteq \mathrm{Range}(\boldsymbol{U}_t)$, and therefore we have $\mathrm{Range}(\boldsymbol{U}_{t-1}) \subseteq \mathrm{Range}(\boldsymbol{U}_t)$. We can write:

$$\begin{aligned}
\|(\boldsymbol{I} - \boldsymbol{U}_t\boldsymbol{U}_t^\top)\boldsymbol{G}_{t-1}(\boldsymbol{I} - \boldsymbol{V}_t\boldsymbol{V}_t^\top)\|_* &\le \|(\boldsymbol{I} - \boldsymbol{U}_t\boldsymbol{U}_t^\top)(\boldsymbol{G}_{t-1} - \hat{\boldsymbol{G}}_{t-1})(\boldsymbol{I} - \boldsymbol{V}_t\boldsymbol{V}_t^\top)\|_* \\
&\quad + \|(\boldsymbol{I} - \boldsymbol{U}_t\boldsymbol{U}_t^\top)\hat{\boldsymbol{G}}_{t-1}(\boldsymbol{I} - \boldsymbol{V}_t\boldsymbol{V}_t^\top)\|_* \\
&\le \|(\boldsymbol{I} - \boldsymbol{U}_t\boldsymbol{U}_t^\top)(\boldsymbol{G}_{t-1} - \hat{\boldsymbol{G}}_{t-1})(\boldsymbol{I} - \boldsymbol{V}_t\boldsymbol{V}_t^\top)\|_* \\
&\le \|(\boldsymbol{I} - \boldsymbol{U}_{t-1}\boldsymbol{U}_{t-1}^\top)\boldsymbol{G}_{t-1}(\boldsymbol{I} - \boldsymbol{V}_{t-1}\boldsymbol{V}_{t-1}^\top)\|_*
\end{aligned} \tag{46}$$

where in the last inequality we leveraged the bound from Theorem 4.3 that upper bounds the tangent space projection error. Particularly, the fact that $\|\boldsymbol{G}_{t-1} - \hat{\boldsymbol{G}}_{t-1}\|_* = \|(\boldsymbol{I} - \boldsymbol{U}_{t-1}\boldsymbol{U}_{t-1}^\top)\boldsymbol{G}_{t-1}(\boldsymbol{I} - \boldsymbol{V}_{t-1}\boldsymbol{V}_{t-1}^\top)\|$. Plugging Equation 46 into Equation 45, we have the following recursive equation:

$$\|(\boldsymbol{I} - \boldsymbol{U}_t\boldsymbol{U}_t^\top)\boldsymbol{G}_t(\boldsymbol{I} - \boldsymbol{V}_t\boldsymbol{V}_t^\top)\|_* \le \|(\boldsymbol{I} - \boldsymbol{U}_{t-1}\boldsymbol{U}_{t-1}^\top)\boldsymbol{G}_{t-1}(\boldsymbol{I} - \boldsymbol{V}_{t-1}\boldsymbol{V}_{t-1}^\top)\|_* + \eta L + \|\boldsymbol{\Delta}_t - \boldsymbol{\Delta}_{t-1}\|_* \tag{47}$$

Unrolling this recursive relation, and considering the SVD initialization at the beginning of Algorithm 1, we have:

$$\|(\boldsymbol{I} - \boldsymbol{U}_t\boldsymbol{U}_t^\top)\boldsymbol{G}_t(\boldsymbol{I} - \boldsymbol{V}_t\boldsymbol{V}_t^\top)\|_* \leq \|(I - \boldsymbol{U}_0\boldsymbol{U}_0^\top)\boldsymbol{G}_0(I - \boldsymbol{V}_0\boldsymbol{V}_0^\top)\|_* + \eta L t + \sum_{i=1}^{t}\|\boldsymbol{\Delta}_t - \boldsymbol{\Delta}_{t-1}\|_* \tag{48}$$

Let $e_0 = \|(I - \boldsymbol{U}_0\boldsymbol{U}_0^\top)\boldsymbol{G}_0(I - \boldsymbol{V}_0\boldsymbol{V}_0^\top)\|_*$. Note that by our initialization of $\boldsymbol{U}_0$ and $\boldsymbol{V}_0$, $e_0$ is expected to be small. For instance, if we assume $\text{rank}(\boldsymbol{G}_0) \leq r$, then we would have $e_0 = 0$. Therefore, for simplicity we can ignore this term.

Plugging Equation 48 back into Equation 44 we have:

$$\mathbb{E}\big[\|\hat{\boldsymbol{M}}_t - \boldsymbol{M}_t\|_*\big] \leq \beta\mathbb{E}\big[\|\hat{\boldsymbol{M}}_{t-1} - \boldsymbol{M}_{t-1}\|_*\big] + \eta L t + \frac{2t\sigma}{\sqrt{B}} \tag{49}$$

Similar to Equation 40, we can unroll the above equation to get:

$$\sum_{t=1}^{T}\mathbb{E}\big[\|\hat{\boldsymbol{M}}_t - \boldsymbol{M}_t\|_*\big] \leq \frac{\eta L T}{1 - \beta} + \frac{2\sigma T}{\sqrt{B}(1 - \beta)} \tag{50}$$

where we also used the fact that $\hat{\boldsymbol{M}}_0 = \boldsymbol{M}_0$. $\qquad\square$

*Proof of Theorem 4.5.* Taking the expectation and unrolling the descent lemma, Lemma D.3, we can write:

$$\begin{aligned}
\frac{1}{T}\sum_{t=0}^{T}\mathbb{E}[\|\bar{\boldsymbol{G}}_t\|_*] &\leq \frac{\mathcal{L}(\boldsymbol{W}_0) - \mathbb{E}[\mathcal{L}(\boldsymbol{W}_{T+1})]}{\eta T} + \frac{2}{T}\sum_{t=0}^{T}\mathbb{E}[\|\hat{\boldsymbol{M}}_t - \bar{\boldsymbol{G}}_t\|_*] + \frac{\eta^2 L}{2} \\
&\leq \frac{\mathcal{L}(\boldsymbol{W}_0) - \mathbb{E}[\mathcal{L}(\boldsymbol{W}_{T+1})]}{\eta T} + \frac{\eta^2 L}{2} + \frac{2}{T}\sum_{t=0}^{T}\mathbb{E}[\|\hat{\boldsymbol{M}}_t - \boldsymbol{M}_t\|] + \frac{2}{T}\sum_{t=0}^{T}\mathbb{E}[\|\boldsymbol{M}_t - \bar{\boldsymbol{G}}_t\|_*]
\end{aligned} \tag{51}$$

Plugging Equation 41 (Lemma D.4) and Equation 50 (Lemma D.5) into Equation 51 yields

$$(1 - \frac{2\beta}{1 - \beta})\frac{1}{T}\sum_{t=0}^{T}\mathbb{E}[\|\bar{\boldsymbol{G}}_t\|_*] \leq \frac{\mathcal{L}(\boldsymbol{W}_0) - \mathbb{E}[\mathcal{L}(\boldsymbol{W}_{T+1})]}{\eta T} + \frac{\eta^2 L}{2} + \frac{4\sigma}{(1 - \beta)\sqrt{B}} + \frac{2\eta L}{1 - \beta} \tag{52}$$

Let $\beta \in (0, \frac{1}{3})$, $\eta < 1$, and $B = T$; then we can rewrite the above equation as follows:

$$\frac{1}{T}\sum_{t=0}^{T}\mathbb{E}[\|\bar{\boldsymbol{G}}_t\|_*] \leq \mathcal{O}\left(\frac{\mathcal{L}(\boldsymbol{W}_0) - \mathbb{E}[\mathcal{L}(\boldsymbol{W}_{T+1})]}{\eta T} + \eta L + \frac{\sigma}{\sqrt{T}}\right) \tag{53}$$

Finally, by letting $\eta = \Theta\left(\sqrt{\frac{\mathcal{L}(\boldsymbol{W}_0) - \mathbb{E}[\mathcal{L}(\boldsymbol{W}_{T+1})]}{TL}}\right)$, we have the final upper bound as follows:

$$\frac{1}{T}\sum_{t=0}^{T}\mathbb{E}[\|\bar{\boldsymbol{G}}_t\|_*] \leq \mathcal{O}\left(\frac{(\mathcal{L}(\boldsymbol{W}_0) - \mathbb{E}[\mathcal{L}(\boldsymbol{W}_{T+1})])^{\frac{1}{2}}\sqrt{L} + \sigma}{\sqrt{T}}\right). \tag{54}$$

$\square$

