# OpenReview forum: "Low-rank Momentum Factorization for Memory Efficient Training"
_TMLR — Accepted by TMLR_

### Review · Reviewer_rb63 · 2025-04-18

**Summary Of Contributions:**

This paper presents MoFaSGD, a novel memory-efficient optimizer that combines the strengths of low-rank subspace methods and adaptive optimization. It introduces a simple yet powerful approach that maintains a low-rank factorization of the momentum and uses it directly for parameter updates, avoiding costly full-rank operations.

Key contributions include:

1. Low-rank momentum representation to reduce memory usage without sacrificing adaptivity.

2. The paper proves theoretical guarantees of optimal convergence rates under standard assumptions.

3. The paper demonstrates strong performance across pretraining and fine-tuning tasks, outperforming other low-rank methods like GaLore and LoRA while retaining similar or better memory efficiency.

In summary, MoFaSGD offers an alternative solution for efficient training of large models.

**Audience:**

Yes

**Claims And Evidence:**

Yes

**Requested Changes:**

1. Add a more detailed analysis or ablation on how the choice of rank affects training stability, convergence speed, and final performance.

2. Incorporate baselines such as signSGD or SWAN to position MoFaSGD more clearly in the landscape of memory-efficient optimizers.

3. Provide more clarity on implementation choices such as how momentum factors are initialized and how projection normalization is handled.

4. Include memory profiling results that break down the contribution of model parameters, optimizer states, and momentum buffers. This would support claims about memory efficiency with more granularity.

**Strengths And Weaknesses:**

## Strengths

1. Provides convergence guarantees and justifies tangent-space projection as optimal among low-rank alternatives.
2. Demonstrates strong results across pretraining and fine-tuning benchmarks, consistently outperforming GaLore and LoRA.
3. Avoids full SVDs and complex second-moment approximations, making the method scalable and easy to implement.

## Weaknesses

1. While multiple ranks are tested, a deeper analysis of how rank impacts stability and performance is missing.
2. Methods like signSGD or SWAN are not included in empirical comparisons.
3. Some implementation choices (e.g., projection normalization, initialization of momentum factors) could be more clearly explained.
4. Profiling results on the memory consumption of trainable parameters and momentum could be useful.

---

> ### Author Response · Authors · 2025-05-23
> **Response to Reviewer rb63**
>
> We are grateful for the reviewer's positive assessment of MoFaSGD's contributions, theoretical guarantees, and empirical results.
>
> 1.  **Deeper Analysis of Rank Impact:**
>     * **Comment:** Need for a more detailed analysis of how rank impacts stability and performance.
>     * **Response:** We have added an **"Ablation: Convergence vs. Efficiency"** subsection in **Section 5.1**. This, along with **Figures 1 & 2** and **Table 1**, provides a more detailed analysis of how rank $r \in \{16, 32, 128\}$ affects convergence speed, final validation loss, and runtime for both MoFaSGD and GaLore during NanoGPT pre-training. The results show MoFaSGD's strong performance, especially at lower ranks.
>
> 2.  **Incorporation of Baselines like signSGD or SWAN:**
>     * **Comment:** Suggestion to include baselines like signSGD or SWAN.
>     * **Response:**
>         * We agree that including these baselines would be ideal. However, SWAN currently lacks stable open-source implementations, and we were unable to reproduce convergence with our own attempt.
>         * For SWAN, we added a discussion in **Section 5.5 ("Implementation Details")** under **"Stateless optimizers."** We note the current lack of open-source implementations and challenges with convergence in our preliminary attempts. As a proxy for memory comparison, we profiled Muon with its momentum buffer disabled (which has close similarity to SWAN), reported as "SWAN" in **Figure 4** and **Figure 11**. We also discuss potential issues with stateless optimizers regarding gradient accumulation, which MoFaSGD handles effectively.
>
> 3.  **Clarity on Implementation Choices (Initialization, Projection Normalization):**
>     * **Comment:** Need for more clarity on momentum factor initialization and projection normalization.
>     * **Response:**
>         * **Initialization:** **Section 5.5 ("Implementation Details")** now has an **"Initialization"** subsection explaining that momentum factors are initialized using the SVD of the first gradient: $(\boldsymbol{U}_{0}, \boldsymbol{\Sigma}_{0}, \boldsymbol{V}_{0}) \leftarrow \text{SVD}_r(\boldsymbol{G}_{0})$, alongside other critical implementation aspects. We also Addressed in Section 5.5 ("Gradient Accumulation and Fused Implementation"), explaining how MoFaSGD handles gradient accumulation with low-rank gradient buffers to maintain memory efficiency.
>
> 4.  **Memory Profiling Breakdown (Parameters, Optimizer States, Momentum):**
>     * **Comment:** Request for more granular memory profiling.
>     * **Response:** This has been significantly enhanced. **Section 5.4 ("Memory Usage Breakdown and Profiling")** and **Appendix C.6** (including **Table 8** and **Figures 9-14**) now provide a detailed breakdown of memory into parameters, optimizer states (which for MoFaSGD include the low-rank momentum factors), gradients, activations, and adapters. This clearly quantifies MoFaSGD's memory footprint.

---

### Review · Reviewer_e2hn · 2025-04-29

**Summary Of Contributions:**

This paper proposes MoFaSGD, a novel low-rank, gradient-based optimizer for deep neural network training, focused on the memory efficiency. The core idea is based on the empirical observation that not only the gradients themselves are low-rank, but their associated momentums also exhibit low-rank structure. By applying an SVD on the gradient and updating the resulting $U, \Sigma, V$ matrices separately (similar to the moving average), MoFaSGD effectively projects gradients into a lower-dimensional “momentum factor” subspace, thereby minimizing the difference in terms of Frobenius-norm. Unlike GaLore, which requires periodic offline updates, MoFaSGD’s fully online updates simplify the algorithm and leads to faster adaptation to training dynamics. Experiments on NanoGPT, RoBERTa, and Llama3.1-8B models with various benchmarks show that MoFaSGD achieves competitive performance compared to existing low-rank based methods.

**Audience:**

Yes

**Broader Impact Concerns:**

I have found no significant broader impact concerns. The paper is proposing a new optimizer which can be used for almost every deep learning models.

**Claims And Evidence:**

Yes

**Requested Changes:**

* Please see the weaknesses above. In particular, consider including the wall-clock time comparisons—plotting Figure 2 with wall-clock time on the x-axis would be acceptable.

**Strengths And Weaknesses:**

**<Strengths>**
* The paper shows a balance between rigorous mathematical derivation and supporting empirical observations. Note that I could not check all the mathematical details and theoretical derivations; but in my perspective, the derivation appears reasonable.
* Thorough comparisons with prior methods such as GaLore, Muon, and LoRA are provided both in the main text and experimentally, clearly demonstrating MoFaSGD’s contributions. In particular, the online updating behavior of MoFaSGD is a notable advantage over GaLore.
* Evaluation across multiple architectures and tasks strengthens the generalizability.

**<Weaknesses>**
* Experimental results mainly compare performance at the same number of training steps or epochs (x-axis). Since the reduced computational complexity does not always translate directly into wall-clock speedups, adding experiments that compare baselines at equal real-time budgets would make the evaluation richer.
* The method relies on the assumption that low-rank gradients yield low-rank momentum. A formal argument of this property would significantly support the paper’s claim.
* In Figure 3(a), the distribution of singular values seems to be broadened as training proceeds. This suggests that, for large-scale LLM training (more data, longer training time), MoFaSGD might show a wider gap vs. full-rank optimizers.
* (minor) Table 2 shows only marginal memory efficiency gains over existing approaches. This is understandable given the small rank r; however, this modest improvement makes this paper more focused to performance and convergence speed rather than the memory efficiency.
* (minor) Technically, LoRA is not an optimizer and introduces additional parameters. Please consider clarifying this distinction in the manuscript.

---

> ### Author Response · Authors · 2025-05-23
> **Response to Reviewer e2hn**
>
> We thank Reviewer e2hn for the positive assessment of our work's balance, comparisons, and generalizability.
>
> 1.  **Wall-Clock Time Comparisons:**
>     * **Comment:** Request for experiments comparing baselines at equal real-time budgets.
>     * **Response:** This has been addressed.
>         * In Section 5.1 (NanoGPT), the new **Figure 2** plots validation loss against wall-clock time. **Table 1** includes runtimes.
>         * In Section 5.2 (Tulu3), **Figure 5b** plots validation loss against wall-clock time, as well as the throughput.
>
> 2.  **Formal Argument for Low-Rank Momentum from Low-Rank Gradients and Broadening Distribution of Singular Values:**
> * **Comment:**
> Request for a formal argument connecting low-rank gradients to low-rank momentum, and an observation that the broadening of the singular value distribution (e.g., in Figure 6a) may indicate a widening performance gap from full-rank optimizers over longer training.
>     * **Response:** Our method is grounded in the empirical observation that the momentum buffer—computed as an exponential moving average (EMA) of gradients—tends to exhibit a low-rank structure. This aligns with prior findings that gradients themselves often lie in a low-dimensional subspace (Zhao et al., 2024a), and that their EMA or second-moment statistics show rapid spectral decay (Feinberg et al., 2024). In Section 5.3 (“Momentum Spectral Analysis”), we provide strong empirical evidence supporting this intuition. Specifically, Figure 6a illustrates that the majority of the momentum’s energy is concentrated in a small number of singular values during Tulu3 instruction tuning, justifying the design of MoFaSGD around low-rank momentum factorization. Nonetheless, we acknowledge that this low-rank modeling introduces inherent limitations. While we reconstruct a full-rank momentum via lossy factor-based approximation, the updates remain restricted to a low-rank subspace. Note that GaLore, for example, operates entirely in this low-rank space without reconstruction. As shown in Table 4, both MoFaSGD and GaLore trail full-rank AdamW by a modest margin (e.g., MoFaSGD is −4.2% avg. on Tulu3), highlighting the trade-off between memory efficiency and full expressivity. The broadening of the singular value distribution over training, as observed in Figure 6a, further indicates that some high-rank structure is inevitably lost. Our spectral analysis supports the view that this is a fundamental shortcoming of current subspace methods—one we believe warrants deeper theoretical investigation and future research. While a formal proof connecting low-rank gradients to low-rank momentum remains outside the scope of this paper, our empirical findings offer a robust justification for the low-rank structure leveraged by MoFaSGD.
>
>
>
> 4.  **Marginal Memory Efficiency Gains in Table 3 (minor):**
>     * **Comment:** Table 2 shows only marginal memory efficiency gains over existing approaches.
>     * **Response:** You've highlighted an important aspect of MoFaSGD's goals. While MoFaSGD does achieve a memory footprint comparable to or slightly better than other low-rank methods like GaLore on certain setups (e.g., RoBERTa-Base in Table 3 where absolute differences are smaller), its primary aim is to enhance convergence speed and sample efficiency over such methods while operating within a similarly efficient memory budget. We show that MoFaSGD consistently achieves lower validation loss than GaLore (Figures 1, 3b, 5) and demonstrates superior sample efficiency and faster wall-clock convergence (Figure 5). The significant memory reduction compared to full-rank AdamW (e.g., MoFaSGD at 29.4 GB vs. AdamW at 70.8 GB for LLaMA-3.1-8B) is a crucial benefit that enables these performance gains on large models under memory constraints. Our detailed memory breakdown for LLaMA-3.1-8B in Section 5.4, Figure 4, and Appendix C.6 (Table 8) contextualizes these memory aspects more broadly.
>
> 5.  **Clarification on LoRA not being an Optimizer (minor):**
>     * **Comment:** Request to clarify that LoRA is not an optimizer.
>     * **Response:** Thank you for catching this. We’ve revised phrasing to explicitly distinguish LoRA as a parameter-efficient fine-tuning method, not an optimizer.

---

### Review · Reviewer_Gyu5 · 2025-05-09

**Summary Of Contributions:**

This paper introduces Momentum Factorized SGD (MoFaSGD), a memory‑efficient optimization method aimed at fine‑tuning large neural networks under tight GPU‑RAM budgets. The authors observe empirically that the first‑order momentum buffer (the exponential moving average of gradients) is itself approximately low‑rank. Leveraging this, they maintain a rank‑r truncated SVD of the momentum at every iteration and update it efficiently using a tangent–space projection that avoids expensive full‑matrix SVDs. The same low‑rank factors $U_{t+1},\Sigma_{t+1},V_{t+1}$ are reused to issue spectrally‑normalized updates $W_{t+1}=W_{t}-\eta\,U_{t+1}V_{t+1}^{\top}$, yielding full‑parameter steps with LoRA‑like memory. Empirically, MoFaSGD is benchmarked against AdamW, Muon, GaLore and LoRA on three regimes: (i) a NanoGPT “speed‑run” pre‑training on 0.73 B tokens; (ii) GLUE fine‑tuning of RoBERTa‑Base; (iii) instruction‑tuning Llama‑3‑8 B on the Tulu‑3 mixture. Across ranks $r\in\{4,8,16,32,128\}$ it attains lower validation loss than GaLore, matches or exceeds LoRA on GLUE and improves over both on Tulu‑3 while requiring comparable memory.

**Audience:**

Yes

**Claims And Evidence:**

Yes

**Requested Changes:**

1. Provide actual peak GPU memory and step‑time measurements on at least one configuration (e.g. Llama‑3‑8 B, r = 8) to substantiate the claimed LoRA‑level efficiency. This is critical for acceptance.
2. Report training throughput (tokens/sec) for MoFaSGD vs AdamW, GaLore and LoRA in the NanoGPT and Tulu‑3 settings. Important but not blocking.
3. Add an adaptive‑rank experiment (e.g. shrinking or expanding r according to a spectral‑energy threshold) to test robustness; alternatively discuss limitations. Desirable.
4. Include sensitivity plots for \beta and learning rate to evidence stability. Desirable.
5. **Important: clarify whether nuclear‑norm smoothness holds for transformer losses or justify the assumption.**
6. The proof of Theorem 4.3 currently sketches the optimality argument.

**Strengths And Weaknesses:**

Strengths:
- The paper tackles a practical problem, which is optimizer RAM overhead during LLM fine‑tuning and proposes a conceptually neat remedy that sits between PEFT and full‑rank adaptation. The factorised momentum idea is original and well‑motived by an empirical spectral study
- The tangent‑space projection is theoretically proved optimal and practically easy to compute via two skinny QR decompositions and one 2r\times2r SVD.
- The convergence proof is carefully written under nuclear‑norm smoothness assumption, matching lower bounds and improving upon prior low‑rank optimisers that lacked guarantees.
- Experiments span pre‑training and alignment‑style fine‑tuning, include strong baselines (Muon, GaLore, LoRA) and demonstrate consistent gains.
- Code availability increases reproducibility.

Weaknesses:
- First, although Table 1 lists memory costs, actual GPU peak memory during training is not reported. This is important because MoFaSGD still stores full‑precision activations and gradients, and the extra QR‑SVD may raise activation checkpointing overhead. Second, computational cost is discussed only asymptotically
- Wall‑clock training times and throughput relative to GaLore and LoRA are missing.
- All experiments adopt fixed ranks, however, I think an adaptive‑rank study would clarify robustness.
- While the theory assumes unbiased gradient oracles and nuclear‑norm smoothness, modern transformers violate the latter. A discussion of practical mismatch should have been provided.

---

> ### Author Response · Authors · 2025-05-23
> **Response to Reviewer Gyu5**
>
> We thank the reviewer for the thoughtful and detailed comments highlighting both the strengths and areas for improvement in our work. Below, we address each point raised:
>
> 1.  **Peak GPU Memory and Step-Time Measurements (Critical):**
>     * **Comment:** Lack of actual GPU peak memory and step-time measurements.
>     * **Response:** We have significantly expanded this aspect.
>         * **Section 5.4 "Memory Usage Breakdown and Profiling"** now details memory efficiency, supported by **Figure 4** (empirical memory breakdown for LLaMA-3.1-8B) and **Figure 7** (GPU memory trace for MoFaSGD).
>         * **Appendix C.6 ("Memory Profiling Details")** provides **Table 8** with a quantitative memory breakdown and **Figures 9-14** with memory traces for MoFaSGD and baselines when training LLaMA-3.1-8B. These show MoFaSGD's memory usage (29.4 GB) is competitive with fused GaLore (30.0 GB) and LoRA (33.6 GB), and much lower than AdamW (70.8 GB).
>         * We also added a discussion on how gradient accumulation is handled with fused implementations to achieve these savings (Section 5.5).
>
> 2.  **Wall-Clock Training Times and Throughput (Important):**
>     * **Comment:** Computational cost discussed only asymptotically; missing wall-clock times and throughput.
>     * **Response:** We have now included these metrics.
>         * For **NanoGPT pre-training** (Section 5.1): **Table 1** reports "Runtime (s)" and "Throughput (tokens/sec)". **Figure 2** plots validation loss against wall-clock time, showing MoFaSGD's runtime efficiency.
>         * For **Tulu3 instruction-tuning** (Section 5.2): **Figure 5b** shows Validation Loss vs. Wall-clock Time, and we also report throughput in "Performance Analysis" paragraph of section 5.2 .
>
> 3.  **Adaptive-Rank Study (Desirable):**
>     * **Comment:** Suggestion for an adaptive-rank experiment or discussion of limitations.
>     * **Response:** We acknowledge that an adaptive-rank mechanism is an interesting research direction. While our current work focuses on fixed ranks, which is a common setting for evaluating such methods, we have added a more detailed ablation on how different fixed ranks affect performance and efficiency in **Section 5.1 "Ablation: Convergence vs. Efficiency,"** supported by **Figures 1 & 2** and **Table 1**.
>
> 4.  **Practical Mismatch of Nuclear-Norm Smoothness & Sensitivity Plots (Desirable):**
>     * **Comment:** Discussion needed on nuclear-norm smoothness violation in transformers; request for sensitivity plots for $\beta$ and learning rate.
>     * **Response:**
>         * **Nuclear-Norm Smoothness:** We have added **Appendix D.1 "On the Choice of Nuclear-norm Smoothness"** to provide a detailed justification for this assumption, drawing connections to modular optimization theory and its relevance to transformer architectures. We also acknowledge the general limitation of non-convex convergence analyses for practical deep learning in our broader discussion of limitations.
>         * **Sensitivity Plots:** We appreciate this suggestion regarding sensitivity plots for β and the learning rate to further evidence stability. For this revision, we had to prioritize addressing other critical feedback points and unfortunately did not have sufficient time to conduct and incorporate a dedicated sensitivity analysis. For the experiments presented, we did perform hyperparameter tuning to identify suitable values for β and the learning rate for each setup, as detailed in Appendix C (e.g., Table 5)
>
>
> 5.  **Proof of Theorem 4.3:**
>     * **Comment:** The proof of Theorem 4.3 currently sketches the optimality argument.
>     * **Response:** We have expanded the proof of Theorem 4.3 in **Appendix D.3** to make it clear. Please let us know if this change actually make it clear.

---

### Author Response · Authors · 2025-05-23
**Authors' Response to Reviews and Manuscript Revision**

We sincerely thank all reviewers for their time and valuable feedback. The suggestions have been instrumental in improving the clarity, rigor, and completeness of our work. We have made significant revisions to the manuscript to address the concerns raised, and have uploaded an updated version, with all additions and changes highlighted in blue. Below, we detail our responses to each reviewer.

---

### Decision · Action_Editor_o4kw · 2025-06-17

**Recommendation:** Accept with minor revision

**Additional Comments:**

## Requested revisions:
- Split long/dense paragraphs for readability, e.g., paragraph 1 in the introduction, **Adaptive methods with Switched off Momentums** paragraph, **Subspace Optimization Methods** paragraph, **Motivation: Low-Rank Momentum Factors** paragraph, etc.
- Define exponential moving averages (EMAs) on page 2 (delete later definition in page 5)
- Can the authors perform a final editing pass to tighten up the writing (as also suggested by Reviewer e2hn)?  Some examples with suggested changes:

Change:
> Now, here is an interesting observation. If we consider the fact that gtg⊤t closely approximates the GaussNewton components of the true Hessian, as rigorously argued in Morwani et al. (2024), it would make sense to switch off covariance momentum (second moment) and consider the following preconditioners based on Adagrad and Shampoo:

to:
> Furthermore, if we consider that gtg⊤t closely approximates the GaussNewton components of the true Hessian (Morwani et al. (2024)), it thus makes sense to switch off covariance momentum (second moment) and consider the following preconditioners based on Adagrad and Shampoo:

Change:
> Hence, we can see that using Pt,2 recovers spectrally normalized updates, which are sometimes referred to as gradient whitening and studied in Bernstein & Newhouse (2024b); Jordan et al. (2024b); Ma et al. (2024).

to:
> Hence, we can see that using Pt,2 recovers spectrally normalized updates, sometimes referred to as gradient whitening (Bernstein & Newhouse, 2024b), (Jordan et al., 2024b), (Ma et al., 2024).

Change:
> Building on these interesting observations, it seems reasonable to conjecture that the gradient momentum Pti=1 β t−iGi enjoys low-rank properties.

to:
> Building on these observations, we conjecture that the gradient momentum Pti=1 βt−iGi exhibits low-rank properties.

Change:
>  Another point of view is that LoRA Hu et al. (2021) is based on the observation that the final fine-tuning update is low-rank, and Galore Zhao et al. (2024a) is built on the observation that the gradient, in general, both in pre-training and fine-tuning, maintains low-rank structure.

to:
> Low-rank structure has also been widely leveraged for LLM fine-tuning, e.g., LoRA (Hu et al., 2021) adapts low-rank matrices during training, while GaLore (Zhao et al., 2024a) leverages the low-rank structure  of gradients.

Change:
> we conduct experiments across three distinct setups: language model pre-training, natural language understanding (NLU) fine-tuning, and large language model instruction-tuning.

to:
> we conduct experiments across three large language modeling setups: pre-training, natural language understanding (NLU) fine-tuning, and instruction-tuning.

- Remove redefinitions after the first time, e.g., "Momentum Factorized SGD (MoFaSGD)"

- Fix the citations to conform to TMLR formatting, i.e., from the formatting instructions:
> When the authors or the publication are included in the sentence, the citation should not be in parenthesis, using \citet{} (as in “See Hinton et al. (2006) for more information.”). Otherwise, the citation should be in parenthesis using \citep{} (as in “Deep learning shows promise to make progress towards AI (Bengio & LeCun, 2007).”).

**Audience:**

Yes

**Audience Explanation:**

Optimizer memory overhead is an important, active problem in LLM training.  The paper proposes an interesting and novel approach to this problem by leveraging approaches from PEFT.

**Claims And Evidence:**

Yes

**Claims Explanation:**

All reviewers agree that the work is interesting and well motivated.  The paper does well to situate its contributions relative to existing work, provide theoretical results for the convergence of MoFaSGD, and compare to existing methods for pre-training, natural language understanding, and instruction-tuning.  The authors also addressed reviewer concerns regarding the limitations/practicality of the theoretical results and additional experiments demonstrating improved wall-clock/memory performance under MoFaSGD.

---

> ### Author Response · Authors · 2025-07-02
> **Camera-Ready Revision Response**
>
> Dear Action Editor and Reviewers,
>
> Thank you for the positive feedback and the recommendation for acceptance. We appreciate your constructive suggestions, which have helped us improve our paper.
>
> We have now completed all requested minor revisions. In the updated manuscript, we have improved readability by splitting dense paragraphs, conducted a comprehensive editing pass to tighten the prose, and ensured that all definitions and citations conform to TMLR’s formatting guidelines.
>
> Thank you again for your help in improving our manuscript.